# Universal toxin-based selection for precise genome engineering in human cells

Songyuan Li [1✉], Nina Akrap[1,12], Silvia Cerboni[2,12], Michelle J. Porritt [1], Sandra Wimberger[1,3], Anders Lundin [1], Carl Möller[1], Mike Firth[4], Euan Gordon[5], Bojana Lazovic[1,6], Aleksandra Sieńska[1], Luna Simona Pane[1], Matthew A. Coelho [7], Giovanni Ciotta[8], Giovanni Pellegrini[9], Marcella Sini[9], Xiufeng Xu[10], Suman Mitra [11], Mohammad Bohlooly-Y[1], Benjamin J. M. Taylor [8], Grzegorz Sienski [1✉] & Marcello Maresca [1✉]

Prokaryotic restriction enzymes, recombinases and Cas proteins are powerful DNA engineering and genome editing tools. However, in many primary cell types, the efficiency of genome editing remains low, impeding the development of gene- and cell-based therapeutic applications. A safe strategy for robust and efficient enrichment of precisely genetically engineered cells is urgently required. Here, we screen for mutations in the receptor for Diphtheria Toxin (DT) which protect human cells from DT. Selection for cells with an edited DT receptor variant enriches for simultaneously introduced, precisely targeted gene modifications at a second independent locus, such as nucleotide substitutions and DNA insertions. Our method enables the rapid generation of a homogenous cell population with bi-allelic integration of a DNA cassette at the selection locus, without clonal isolation. Toxin-based selection works in both cancer-transformed and non-transformed cells, including human induced pluripotent stem cells and human primary T-lymphocytes, as well as it is applicable also in vivo, in mice with humanized liver. This work represents a flexible, precise, and efficient selection strategy to engineer cells using CRISPR-Cas and base editing systems.

[1] Translational Genomics, Discovery Sciences, BioPharmaceuticals R&D, AstraZeneca, Gothenburg, Sweden. [2] Translational Science and Experimental Medicine, Respiratory & Immunology, BioPharmaceuticals R&D, AstraZeneca, Gothenburg, Sweden. [3] Department of Chemistry & Molecular Biology, University of Gothenburg, Gothenburg, Sweden. [4] R&D Data Infrastructure & Tools, AstraZeneca, Cambridge, UK. [5] Discovery Biology SWE, Discovery Sciences, BioPharmaceuticals R&D, AstraZeneca, Gothenburg, Sweden. [6] Oulu Center for Cell-Matrix Research, Biocenter Oulu and Faculty of Biochemistry and Molecular Medicine, University of Oulu, Oulu, Finland. [7] Wellcome Sanger Institute, Cambridge, UK. [8] Discovery Biology UK, Discovery Sciences, BioPharmaceuticals R&D, AstraZeneca, Cambridge, UK. [9] CVRM pathology, Clinical Pharmacology & Safety Sciences, BioPharmaceuticals R&D, AstraZeneca, Gothenburg, Sweden. [10] Department of Biosciences and Nutrition, Karolinska Institute, Stockholm, Sweden. [11] Inserm UMR1277 CNRS UMR9020 – CANTHER, Institut pour la Recherche sur le Cancer de Lille, Lille, France. [12] These authors contributed equally: Nina Akrap, Silvia Cerboni. ✉email: songyuan.li@astrazeneca.com; grzegorz.sienski@astrazeneca.com; Marcello.Maresca@astrazeneca.com

Gene and cell therapy offer new modalities in medicine for a wide range of diseases[1–4]. Despite the potential of such therapies, technical challenges—such as the precision and low efficiency of current genomic engineering tools[5,6]—restrict their development and clinical application. While the accuracy of genome editing and the ability to predict off-target effects has been considerably improved, the efficiency of genetic engineering in somatic cells, especially precise substitutions and gene insertions, remains generally low, limiting potential therapeutic applications[7–11]. Therefore, there is an urgent need for an approach that specifically selects only those cells with the desired genomic modification.

Cells proficient for genomic editing at one locus are more likely to be proficient for editing events at another locus[12,13]. This principle provides the basis for co-selection of genomic modifications at the desired locus, in combination with an external or endogenous selection marker. Editing of a selection marker can produce a detectible signal or a growth advantage for the edited cell. Cells that undergo simultaneous editing at the selection locus and a second-site targeted locus are typically enriched by fluorescent reporter-based sorting or resistance to specific cytotoxic reagents[14–18]. This co-selection strategy has been used in cells from organisms ranging from Caenorhabditis elegans to humans[19–23]. However, most previous methods involved introducing a random set of insertions or deletions (indels), and few studies involved co-selection of precise DNA substitutions. These, in turn, often introduced unsupervised and risky modifications to the engineered cells[21,24]. A selection method that (1) does not require an external selection marker, (2) specifically eliminates non-edited cells without side effects on edited cells, and (3) introduces precise (safe) modification at the selection locus is still lacking.

Bacterial toxins have high selectivity and potency in eliminating plant and animal cells[25]. Unlike small molecules that freely diffuse through membranes, penetrating all cells, most bacterial toxins are large molecules that enter cells via a specific receptor. Typically, such bacterial toxins consist of two domains: one domain recognizes a specific membrane receptor and mediates endocytosis and translocation, and the second domain executes cytotoxic functions inside the targeted cell[26–28]. This modular structure allows the uncoupling of specificity from toxicity[29]. The diphtheria toxin (DT) from Corynebacterium diphtheriae[30] is composed of domain B (DT-B) that binds to the membrane-embedded form of heparin-binding EGF-like growth factor (HBEGF), and mediates endocytosis and translocation of DT. Once DT enters the cytoplasm, domain A (DT-A) inactivates translation elongation factor 2, causing cell death[30]. DT exhibits toxicity in most mammalian species, with the exception of mice and rats[31]. The resistance of these organism stems from impaired DT binding to their HBEGF homologs, due to differences in amino acid sequence[32]. Introduction of analogous amino acid substitutions from mouse to human HBEGF (hHBEGF) prevents its binding to DT and establishes resistance to the toxin[33].

In this work, we exploit the interaction between DT and HBEGF to develop a universal selection strategy that depletes only those cells that have not introduced the desired genome modifications. Our approach specifically protects the engineered cells, blocking the entry of the lethal toxic molecule. We find that by introducing DT-resistant mutations into HBEGF and selecting for edited cells with DT, we observe a substantial increase in a simultaneous, second-site gene editing event in these cells. This principle holds true for a variety of genome engineering events mediated by DNA base editors and Cas9 nuclease, including HBEGF locus-specific biallelic insertion of a DNA cassette. Finally, we demonstrate that our DT-HBEGF selection system is applicable both in vitro, in therapeutically relevant cell types, such as human-inducible pluripotent stem cells (hiPSCs) and primary human T cells, as well as in vivo in mice with a humanized liver.

## Results

### HBEGF locus mutagenesis generating diphtheria toxin resistance.
DT interacts with the EFG-like domain of HBEGF[2]. Replacement of the mouse EGF-like domain with corresponding human domain causes mouse cells to become sensitive to DT[2]. To induce mutations in the human EGF-like domain that would render human cells insensitive to DT, we used the DNA base editors cytidine base editor 3 (CBE3) and adenosine base editor 7.10 (ABE7.10; Fig. 1a)[34,35]. We designed 14 sgRNAs spanning the amino acids that differ between mouse and human at this locus (Fig. 1b). Each sgRNA was transiently co-expressed in HEK293 cells with either CBE3 or ABE7.10 to introduce C-to-T or A-to-G mutations, respectively[34,35]. The transfected cells were treated with a DT dose that elicits cell death unless the interaction with HBEGF has been disrupted. We monitored cell proliferation and observed that cells transfected with CBE3 and sgRNA7 or 10, and ABE7.10 with sgRNA5 or 6 continued to proliferate despite the presence of the toxin in the cell culture medium (Fig. 1c). The cells transfected with other combinations of plasmids, as well as the control cells did not survive the treatment (Fig. 1c).

We chose resistant cells transfected with CBE3/sgRNA10 or ABE7.10/sgRNA5, and sequenced the targeted HBEGF locus (Fig. 1d). Mutations introduced by CBE3/sgRNA10 resulted mainly in the substitution of glutamate 141 for lysine (Glu141Lys) in HBEGF, whereas ~90% of the mutations introduced by the ABE7.10/sgRNA5 combination elicited a substitution of tyrosine 123 to cysteine (Tyr123Cys; Fig. 1d and Supplementary Fig. 1a). HBEGF Glu141 plays a key role in the DT-HBEGF interaction and the Glu141His substitution abolishes DT sensitivity, when expressed in mouse cells (Supplementary Fig.1a)[36–38]. HEK293 cells edited to express HBEGF[Glu141Lys] or HBEGF[Tyr123Cys] show wild-type levels of proliferation (Supplementary Fig.1c). We detected noticeable levels of indels induced by CBE3 and to a lesser extent with ABE7.10 (Supplementary Fig.1b), as previously observed[34,39]. Thus, the substitution of a single amino acid in hHBEGF is sufficient to prevent DT toxicity, suggesting this method could be used to select for genome editing events at the HBEGF locus.

### Enrichment of cytidine and adenosine base editing using DT selection.
Our approach selects for the survival of cells that are proficient for base editing. We therefore asked if such selection favors simultaneous base editing at another, unrelated genomic locus (co-selection[40]; Fig. 2a). Using CBE3/sgRNA10, we tested for co-selection with sgRNAs targeting five independent genomic loci: DPM2 (dolichyl-phosphate mannosyltransferase subunit 2), EGFR (epidermal growth factor receptor), EMX1 (empty spiracles homeobox 1), PCSK9 (proprotein convertase subtilisin/kexin type 9), and DNMT3B (DNA methyltransferase 3 beta). The sgRNAs targeting DPM2 and PCSK9 were designed to introduce a premature stop codon[8], and the sgRNA targeting EGFR was designed to generate a drug-resistant mutation in EGFR[15]. After co-transfection, we extracted genomic DNA from cells either exposed or not exposed to DT and analyzed the sgRNA-targeted DNA sequence composition. We observed a substantial increase in the C–T conversion rate across all tested sites in DT-selected cells (~4–7-fold), compared to nonselected cells (Fig. 2b). DT co-selection with CBE3 in other cancer cell lines also yielded increased editing efficiency. Using our strategy, we obtained a ~13-fold increase in the C–T substitution rate at the PCSK9 locus in HCT116 cells and a ~5-fold increase at the integrated BFP transgene in PC9 cells (Supplementary Fig. 2). We subsequently

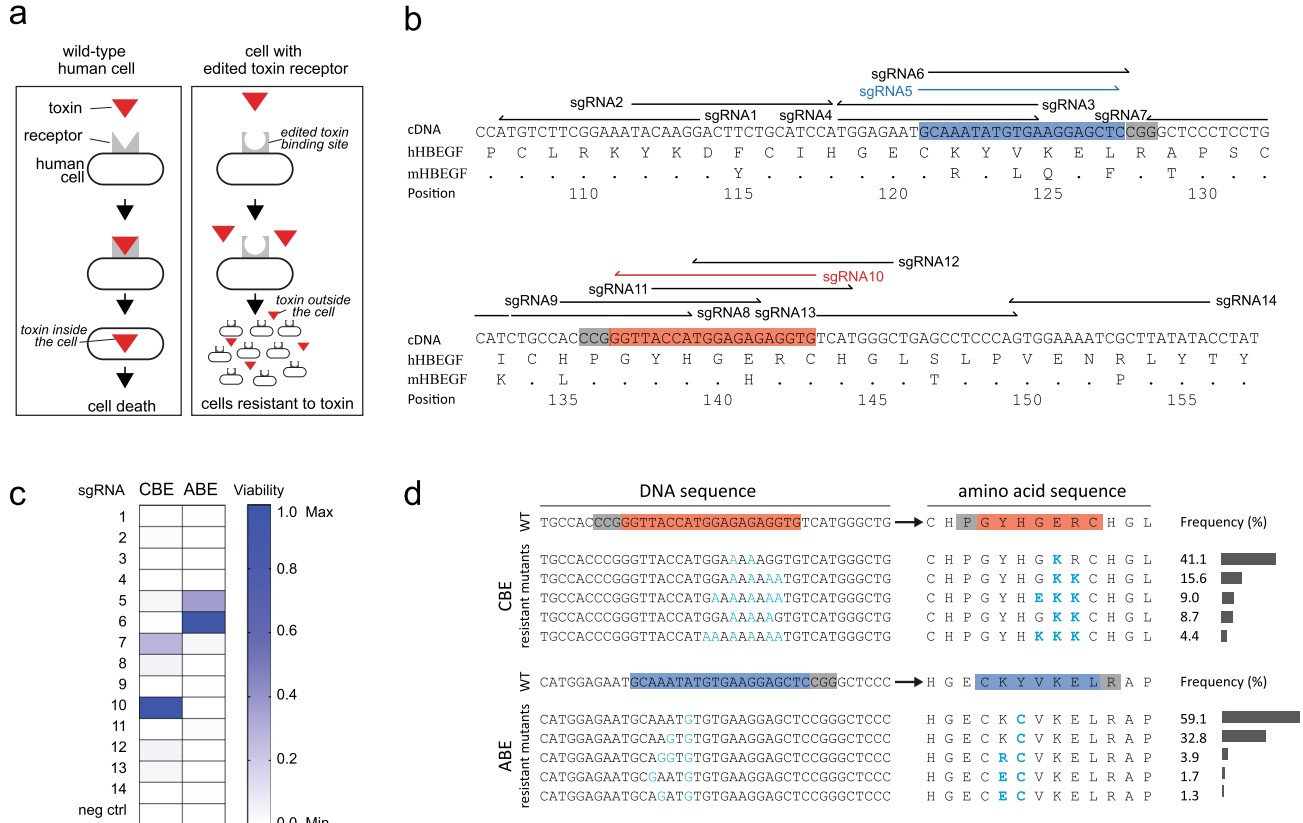

**Fig. 1 Base editing at the *HBEGF* locus induces resistance to diphtheria toxin. a** Schematic of our toxin-based selection scheme. **b** sgRNA sites targeted by CBE3 or ABE7.10, and used to screen for mutations in *HBEGF* that elicit resistance to DT. cDNA is the DNA sequence of the EGF-like domain of human HBEGF; hHBEGF is the corresponding amino acid sequence; mHBEGF is the aligned amino acid sequence of the mouse HBEGF homolog. Matching amino acids in mHBEGF are shown by a dot; unmatched amino acids are annotated. sgRNAs highlighted in red and blue were chosen to introduce DT-resistant mutations with CBE3 and ABE7.10, respectively. **c** Heatmap presenting the viability of HEK293 cells after DT selection for the depicted combination of base editors and sgRNAs. **d** Frequency of alleles in DT-resistant cells after CBE or ABE editing. The values in **c** and **d** represent an average of three independent biological replicates. CBE, cytidine base editor, ABE adenosine base editor.

tested if DT co-selection applies to the latest version of CBE, CBE4max[41], and found a significant improvement in C–T conversion across three tested targets in DT-selected cells (~4–7-fold; Supplementary Fig. 3a).

To determine if ABE7.10/sgRNA5 editing also promotes co-selection, we tested it with five sgRNAs targeting: *EMX1, CTLA4* (cytotoxic T-lymphocyte-associated protein 4), *IL2RA* (interleukin 2 receptor subunit alpha), and two different sites in the *AAVS1* locus (adeno-associated virus integration site 1). Analyzing the DNA sequences targeted by the sgRNAs with or without the DT treatment, we observed a substantial increase in A–G conversion across all tested targets in DT-selected cells, ranging from ~6 to ~13-fold (Fig. 2c). DT co-selection with the latest ABE version ABE8e (ref. [42]) further improved the editing efficiency (~2-fold; Supplementary Fig. 3b).

We asked if our approach was able to co-select for genome modifications, such as insertions and deletions (indels), generated by the *Streptococcus pyogenes* Cas9 nuclease (SpCas9). We tested whether SpCas9 guided by sgRNA10 to introduce indels in the *HBEGF* locus would promote co-selection, using the four sgRNAs targeting *DPM2, EMX1, PCSK9*, and *DNMT3B* (above). Transfected cells subjected to DT treatment showed increased indel rates (>90%) at all four targets (*DPM2, EMX1, PCSK9*, and *DNMT3B*; Fig. 2d). Thus, DT-HBEGF selection is able to enrich for a range of genome editing events without the need for an external selection marker.

**Enrichment of DNA insertion at the *HBEGF* locus**. A major limitation of Cas9-mediated genomic engineering is the low efficiency with which targeted DNA insertion (DNA knock-in) is generated through homology directed repair (HDR)[43–47] and non-homologous end-joining (NHEJ)[46,48–50]. To test whether our DT-HBEGF system could select for genomic DNA insertions without the use of an external selection marker[40], we modified our DT selection strategy. Our idea was to engineer a DNA template for the targeted insertion in the intron 3 of the *HBEGF* gene. SpCas9 nuclease programmed with sgRNAin3 to target the intron 3 would confer resistance to DT only if the DNA template was integrated at this site, as it contains a splicing acceptor sequence followed by the *HBEGF* cDNA with all exons downstream of exon 3, together with a mutation that confers insensitivity to DT (Fig. 3a). In this modified DT selection strategy, we used the Glu141Lys substitution in the *HBEGF* cDNA (Fig. 1d).

Because the soluble form of HBEGF acts as a ligand for EGFRs, activating downstream signal pathways[51], we therefore determined whether the HBEGF^Glu141Lys protein might perturb this signaling. We purified soluble, recombinant wild-type HBEGF and HBEGF^Glu141Lys from bacteria (Supplementary Fig. 4a), and assayed their ability to activate the mitogen-activated protein kinase (MAPK)/extracellular signal-regulated kinase signaling pathway downstream of EGFR in serum-starved HEK293 cells. The addition of either wild-type or mutant recombinant protein to the cell culture medium resulted in similar levels of signaling,

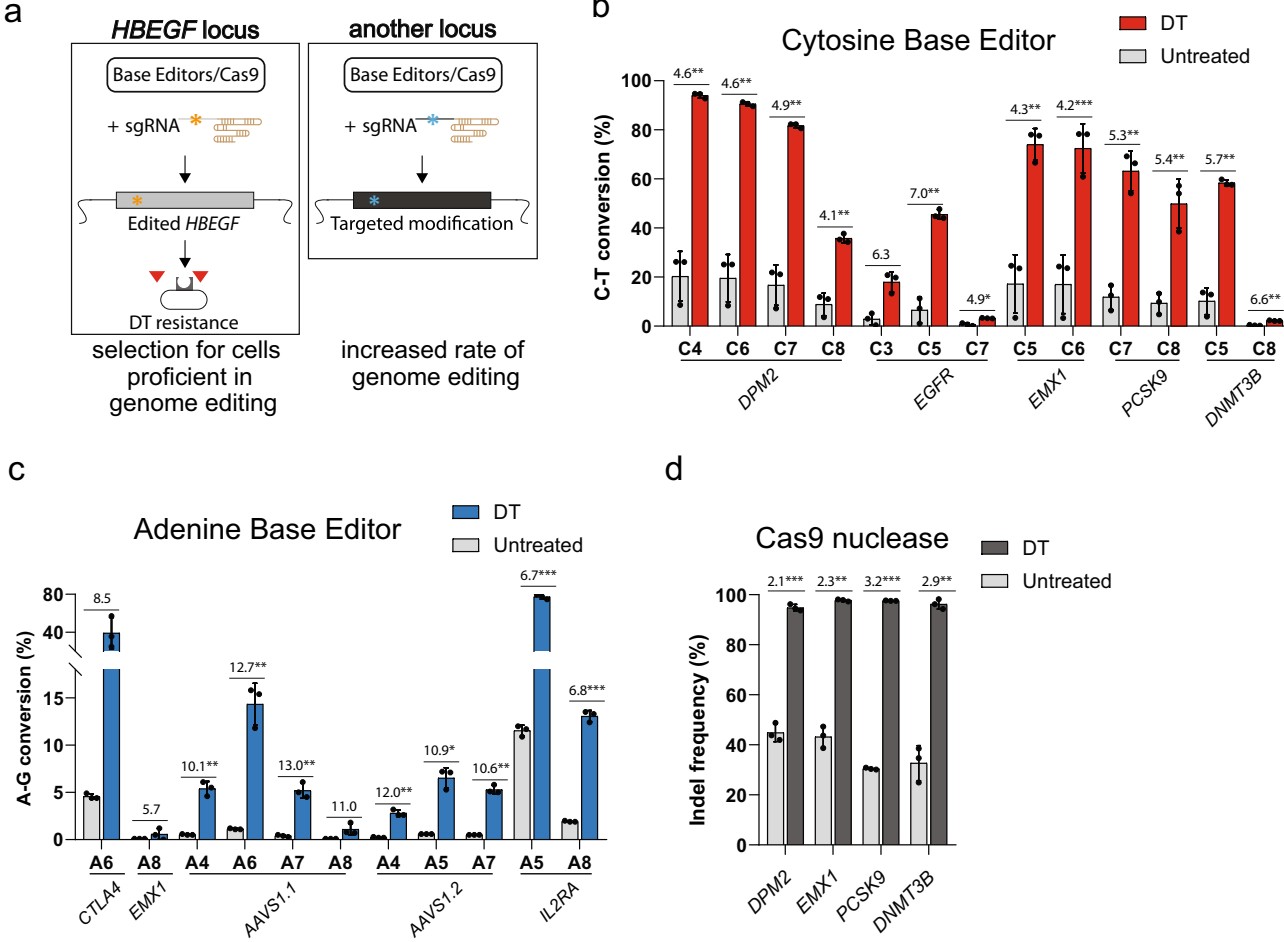

**Fig. 2 Co-selection of base editing- and Cas9 nuclease-mediated genome modifications. a** Schematic of experimental design. **b** Bar graph of co-selected cytidine base editing events at the *HBEGF* locus and another locus, with or without DT selection, showing C–T conversion (%). **c** Bar graph of co-selected adenosine base editing events with or without DT selection, showing A–G conversion (%). **d** Bar graph of co-selected SpCas9-mediated genome editing events. In all graphs, the values and error bars reflect mean ± s.d. of $n = 3$ independent biological replicates. Relative fold changes between DT-selected and nonselected cells are indicated in the graphs. *$P < 0.05$, **$P < 0.01$, ***$P < 0.001$, Student's paired $t$ test (two-tailed). $P$ values are calculated as below: in **b**, *DPM2* (C4 = 0.0070, C6 = 0.0063, C7 = 0.0061, and C8 = 0.0061), *EGFR* (C3 = 0.0506, C5 = 0.0053, and C7 = 0.0184), *EMX1* (C5 = 0.0031 and C6 = 0.0004), *PSCK9* (C7 = 0.0028 and C8 = 0.0092), and *DPM2* (C5 = 0.0038 and C8 = 0.0088); in **c**, *CTLA4* (A6 = 0.0691), *EMX1* (A8 = 0.3119), *AAVS1.1* (A4 = 0.0091, A6 = 0.0090, A7 = 0.0090, and A8 = 0.1044), *AAVS1.2* (A4 = 0.0072, A5 = 0.0107, and A7 = 0.0027), and *IL2RA* (A5 = 0.0006 and A8 = 0.0009); in **d**, *DPM2* (0.0005), *EMX1* (0.0016), *PCSK9* (0.0002), and *DNMT3B* (0.0085). Source data of Fig. 2b–d are provided as a Source data file.

as monitored by phosphorylation of MAPK (Supplementary Fig. 4b)[51], indicating that HBEGF^Glu141Lys similarly to wild-type HBEGF.

We next designed DNA fragments that would serve as DNA repair templates to introduce (a) a DT-resistant *HBEGF* and (b) couple its expression to the expression of a reporter gene encoding a red (mCherry) or green fluorescent protein (GFP) using self-cleaving peptides[52]. We assayed different DNA template donors for targeted insertion, including plasmid DNA, double-strand DNA (dsDNA), and single-strand DNA (ssDNA). We also examined the main modes of DNA repair that direct the DNA insertion, because we used DNA fragments with or without homology arms or flanking sgRNA cutting sites. This design allows us to promote integration into *HBEGF* locus by homologous recombination (HR)[43,47,53], NHEJ[50], or homology-mediated end joining (HMEJ)[54] (Fig. 3b). Each template was co-transfected with plasmids encoding SpCas9 and sgRNAin3 into HEK293 cells to generate cells with the inserted DNA fragment in intron 3 of the *HBEGF* gene. Since the expression of mCherry or GFP is coupled to the *HBEGF* gene, we expected only cells with the correct insertions to express fluorescent proteins. We

therefore quantified fluorescent cells by flow cytometry. The number of the mCherry- or the GFP-positive cells increased substantially after DT selection in all experimental variants tested, regardless of the DNA template used (Fig. 3b). In particular, the plasmid template containing homology arms and sgRNA cutting sites (pHMEJ) or the plasmid template containing only homology arms (pHR) achieved nearly 100% of knock-in after the DT selection (Fig. 3b). We compared the precision of the DNA knock-in that were derived from pHR or pHMEJ. Genotyping strategy with PCR confirmed a dominant band corresponding to HDR repair in both samples (Supplementary Fig. 5). For pHMEJ specifically, we detected an additional, faint band indicating NHEJ-mediated repair (Supplementary Fig. 5).

Cells resistant to DT after base editing showed biallelic mutation at the *HBEGF* locus (Fig. 1b). We therefore reasoned that cells surviving DT treatment have the biallelic insertion of the DNA cassette, given that one intact *HBEGF* allele would sensitize cells to the toxin. In order to test this hypothesis, we designed two pairs of PCR primers, one pair amplifying the 5′ junction of the knock-in sequence (PCR1) and the other pair amplifying the wild-type sequence of *HBEGF* intron 3 (PCR2;

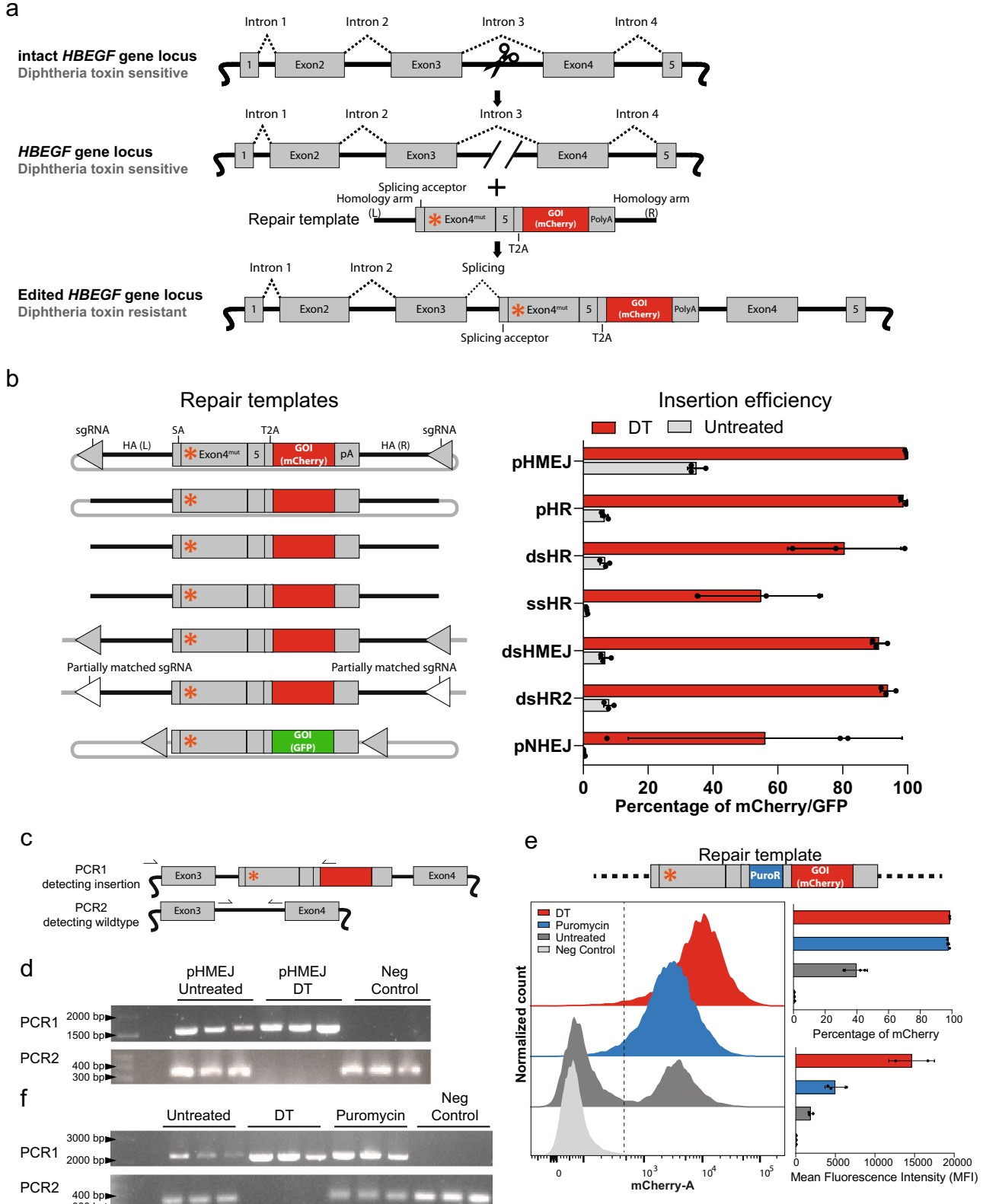

Fig. 3c). We performed the PCR analysis on cells repaired with the pHMEJ template either with or without DT selection. Both samples showed a band for homologous knock-in (PCR1); however, we only detected the wild-type band in the nonselected samples, and not the DT-selected samples (Fig. 3d), indicating that cells showed biallelic knock-in after DT selection. We also analyzed the cells using flow cytometry and measured the

fluorescence of mCherry. DT selection yielded a highly homogenous population of mCherry-positive cells, unlike the non-selected control (Supplementary Fig. 4c).

We compared our DT selection method to antibiotic-based selection methods used to enrich for cells with genomic integration of a transgene. We designed an alternative pHMEJ template that included DT-resistant HBEGF$^{\text{Glu141Lys}}$, a

**Fig. 3 Enrichment of DNA knock-in at the HBEGF locus. a** Schematic of the knock-in enrichment strategy. **b** The knock-in of various templates (left) and their corresponding efficiencies (right). The mCherry/GFP percentage of each sample was analyzed by flow cytometry with (DT) or without (untreated) DT selection. Repair templates were designed to be incorporated into the targeted site through homology-mediated end joining (pHMEJ and dsHMEJ), homologous recombination (pHR, dsHR, ssHR, and dsHR2), or nonhomologous end joining (pNHEJ). These templates were provided as plasmids (pHMEJ, pHR, or pNHEJ), double-stranded DNA (dsHR, dsHMEJ, and dsHR2), or single-stranded DNA (ssHR). **c** Schematic of the genotyping strategy. The PCR1 primer pair detect the insertion; PCR2 detects wild-type cells in the population. **d** PCR analysis of cell populations obtained from experiment (b), representative results were shown from three independent biological replicates. **e** Comparison of puromycin and DT-enriched knock-in populations. Upper panel: the repair template consists of a puromycin resistant gene and a mCherry gene linked to the mutated *HBEGF* gene. The lower left panel shows the representative mCherry histogram of edited HEK293 cell populations without or with different treatments. Neg control represents cells transfected with control sgRNA (no target loci), instead of sgRNAin3. Cells were analyzed by flow cytometry. The lower right panels show corresponding knock-in efficiencies and mean fluorescence intensities of each population. **f** PCR analysis of each population of cells obtained from experiment (e), representative results were shown from three independent biological replicates. The presented values and error bars reflect mean ± s.d. of $n = 2$ or 3 independent biological replicates. GOI gene of interest, T2A T2A self-cleaving peptide, SA splicing acceptor, HA homology arm, pA polyA sequence. Source data of Fig. 3b, d, e, f are provided as a Source data file.

commonly used puromycin-resistance gene and mCherry (Fig. 3e). We tested this DNA template for genomic insertion by selecting cells with either DT or puromycin, followed by flow cytometry analysis. Both populations consisted of nearly 100% mCherry-positive cells, however, DT-selected cells showed a substantially higher mean fluorescence intensity compared to puromycin-enriched cells (Fig. 3e). Genotyping the *HBEGF* locus for the presence of the transgene insertion revealed that DT selection resulted in biallelic insertion (Fig. 3f), unlike the puromycin-selected cells, which contained monoallelic insertions.

Collectively, these data demonstrate that the DT-HBEGF system enables the selection of precise genomic integration events and that the *HBEGF* locus constitutes a selectable locus for such biallelic genomic integration. With only minimal modifications introduced to the human genome, the DT-HBEGF selection system provides an efficient alternative for the generation of a genetically homogenous population of cells. We name the system "Xential" for "recombination (X) at a locus conditionally essential for cell survival" hereafter.

**Enrichment of knock-out and knock-in by Xential.** Several recent studies have demonstrated co-selection of CRISPR-Cas9 nuclease editing events using endogenous genes[14,21]. However, they rely on introducing random mutations in the gene used for enrichment, yielding a heterogenous population of cells, raising safety concerns if applied in a therapeutic setting. Because Xential should generate a homogeneous population of cells with a precise insertion in *HBEGF* intron 3, we reasoned that it could also provide an alternative to co-select for both knock-out and knock-in events at another genomic locus. To test this idea, we assayed the enrichment of Xential-coupled knock-out events, using the four sgRNAs targeting *DPM2*, *EMX1*, *PCSK9*, and *DNMT3B* (Fig. 2d). Each sgRNA was co-transfected with SpCas9, sgRNAin3, and the pHMEJ template into HEK293 cells, followed by DT selection and subsequent genomic DNA analysis (Fig. 4a). We observed a substantial improvement in the editing efficiency for all targets in the DT-selected cells compared to the non-selected cells (~4–14-fold; Fig. 4b). All DT-resistant cells maintained the expression of mCherry (Supplementary Fig. 6a).

To test the use of Xential for co-selection of knock-in events at other genomic loci (Fig. 4c), we introduced a C-terminal *GFP* tag into a gene encoding histone protein H2B (*HIST1H2BC*). We designed pHR and pHMEJ DNA repair templates for both *HBEGF* and *HIST1H2BC* loci, and co-transfected each of them separately with SpCas9 and sgRNAs into HEK293 cells. The knock-in efficiency was analyzed using flow cytometry by calculating the percentage of GFP (*HIST1H2BC-GFP*) or mCherry (*HBEGF-2a-mCherry*). Regardless of the DNA template donor, we obtained substantially improved knock-in efficiency after DT selection (Fig. 4d). For the pHR template, Xential

improved co-selection efficiency sixfold. For the pHMEJ template, the efficiency increased up to ~5-fold, reaching ~50% overall (Fig. 4d). By increasing the ratio of the sgRNA and DNA template for tagging the *HIST1H2BC* locus to that of editing *HBEGF* locus, we were able to further improve the efficiency of knock-in after DT selection (Fig. 4d and Supplementary Fig. 6b), suggests that knock-in efficiency at targeted locus mediated by Xential is dose-dependent.

We next investigated if the coedited cells are more prone to genomic translocation. To this end, we employed a droplet-digital PCR (ddPCR) assay designed to detect the monocentric translocation from the *HIST1H2BC* to the *HBEGF* locus. When only two sgRNAs were used, we observed an increase in genomic translocation in-between the loci. Interestingly, a combination of the two sgRNAs with corresponding DNA repair templates (Xential) vastly suppressed the translocations. This data suggests that the Xential co-selection reduces genomic rearrangements, therefore, provides a safety advantage over previous indels-based co-selection systems (Supplementary Fig. 6c).

To determine if Xential could be used to co-select for the insertion of small DNA fragments, such as oligonucleotides, we tested Xential co-selection for knock-in of a DNA oligo at the *CD34* locus. We observed an increase in the percentage of knock-in cells after DT selection (>35-fold), suggesting Xential improves the integration of DNA cassettes regardless of their size and form (Supplementary Fig. 6d). Thus, Xential promotes precise DT-resistant modification of the *HBEGF* locus, allowing the introduction of two genes of interest simultaneously, or the introduction of one gene of interest together with a second gene knock-out event.

**Enrichment of base editing and DNA insertions in human iPSCs.** Having demonstrated the effectiveness of Xential co-selection in HEK293 cells, we asked if it works equally well in non-cancer-transformed cells. We chose hiPSC, because of their relevance to disease modeling, their therapeutic potential, and the difficulty of genome manipulation[55–57]. We used two sgRNAs for co-selection with CBE and ABE, one targeting *EMX1*, a locus widely tested in genome editing research, and the other targeting *CTLA4*, a gene studied for its role in immune signaling[58]. We optimized the experimental timeline for hiPSC, shortened it to 7 days from transfection to the derivation of DT-resistant cells (Fig. 5a). We transiently co-transfected the sgRNAs together with CBE3/sgRNA10 or ABE7.10/sgRNA5 into hiPSCs, and analyzed the targeted genomic DNA sequence. We observed a substantial increase in editing efficiency upon DT selection (greater than ~20-fold) at all tested sites for both CBE and ABE. For example, the hiPSCs resistant to DT showed ~90% CBE3 modified reads at the *EMX1* locus (from 5% for control cells) and ~20% ABE7.10 modified reads at the *CTLA4* locus (0.8% in control cells; Fig. 5b,

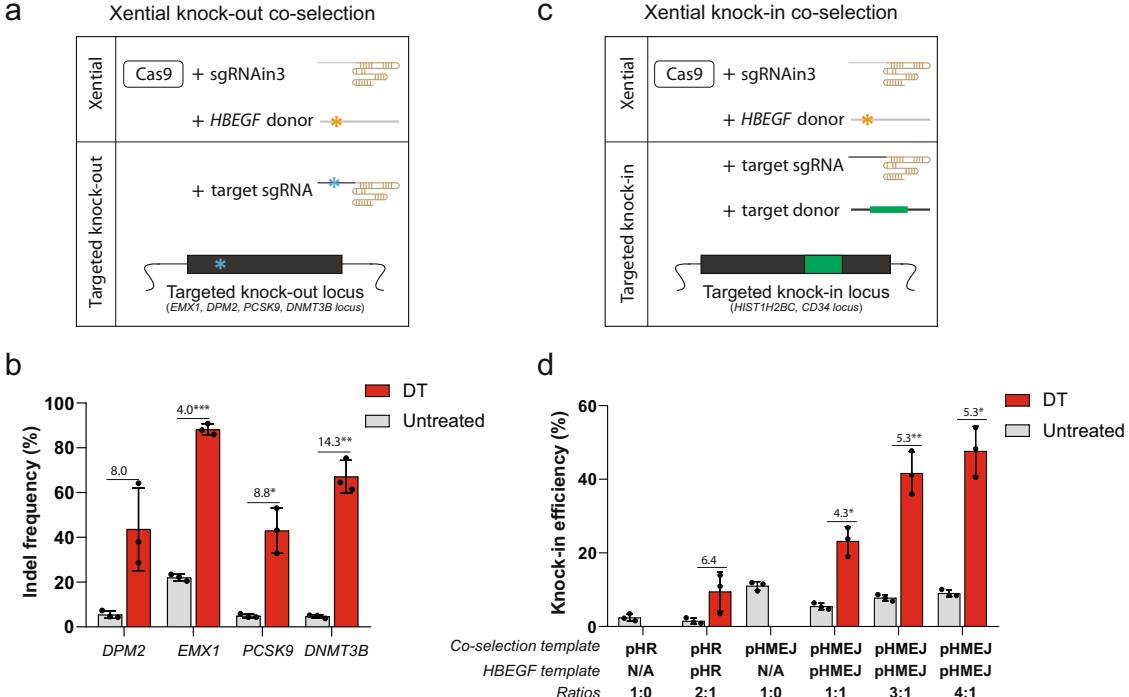

**Fig. 4 Xential enables co-selection of knock-out and knock-in editing. a** Strategy for co-selecting knock-out events with precise knock-in at the *HBEGF* locus. **b** Co-selection of SpCas9 indels in HEK293 cells. Cells were co-transfected with SpCas9, sgRNAin3, the pHMEJ repair template for *HBEGF* locus, and a sgRNA targeting a second genomic locus. Cells were cultivated with (DT) or without DT (Untreated) from 72 h after transfection until confluent. Genomic DNA were extracted from harvested cells and analyzed by NGS. **c** Strategy of co-selecting knock-in events with precise knock-in at the *HBEGF* locus. **d** Co-selection of knock-in events at a second locus, *HIST1H2BC*, in HEK293 cells. Cells were co-transfected with SpCas9, sgRNAs, and repair templates for both *HBEGF* and *HIST1H2BC* locus; cultivated with (DT) or without (Untreated) DT; analyzed by flow cytometry. Both pHR and pHMEJ templates were used. Ratios of the amount of sgRNA and template for *HBEGF* locus to that for *HIST1H2BC* locus are indicated below. N/A indicates no corresponding component was used. Values and error bars reflect mean ± s.d. of $n = 3$ independent biological replicates. Relative fold changes between untreated and DT-treated samples are indicated in the graphs. $*P < 0.05$, $**P < 0.01$, $***P < 0.001$, Student's paired $t$ test (two-tailed). $P$ values are calculated as below: in **b**, *DPM2* (0.0595), *EMX1* (0.0003), *PCSK9* (0.0228), and *DNMT3B* (0.0043); in **d**, $P$ values from left to right are 0.1534, 0.0181, 0.0089, and 0.0128, respectively. Source data of Fig. 4b, d are provided as a Source data file.

c). Similar to our experiments in cancer cells, DT co-selection further improved the efficiency of the latest base editors CBE4-max and ABE8e also in hiPSC (>74% modified reads for all tested sites, Supplementary Fig. 7a, b).

To test if Xential facilitates DNA insertion in the genome of hiPSCs, we co-transfected hiPSCs with SpCas9, sgRNAin3, and the pHMEJ template containing the DT-resistant *HBEGF* variant linked to mCherry. Flow cytometry revealed that ~25% of cells were mCherry positive in the absence of selection. This number increased to nearly 100% after DT selection (Fig. 5d). We corroborated this data using genotyping PCR to detect the DNA insertion and the wild-type *HBEGF* intron 3 (Fig. 5e). We could not detect any residual wild-type band in the targeted *HBEGF* after DT selection, suggesting most of the selected pool of hiPSCs contain biallelic insertions of the DNA cassette (Fig. 5e). All sgRNAs used for DT selection (sgRNA5, sgRNA10, and sgRNAin3) were analyzed in silico and in vitro for off-target sites in hiPSC (Supplementary Table 1). We detected less than 0.1% of modifications (0.1% as NGS detection of limit)[59] at all selected sites that confirmed specificity of all sgRNAs used for co-selection (Supplementary Fig. 8).

To address the impact of the HBEGF^{E141K} on the differentiation of hiPSC, we differentiated the Xential-modified hiPSC to three germ layers: mesoderm, endoderm, and ectoderm. We monitored the expression of key pluripotency- and lineage-associated genes[60], and did not observe any statistically significant

changes between the Xential hiPSC and the wild-type hiPSC (Supplementary Fig. 9).

Finally, we set out to demonstrate the translational potential of our Xential method. To this end, we chose to install a safety switch with Xential in hiPSC[61]. We engineered the pHMEJ plasmid to encode thymidine kinase from Herpes Simplex Virus (HSV-TK), and inserted it with Xential method to hiPSC. After DT selection and cell expansion, we tested the sensitivity of the cell pool to ganciclovir—a synthetic substrate for the viral TK that ultimately inhibits DNA replication[61]. The cells expressing HSV-TK did not survive the ganciclovir treatment (>1 µM), unlike the control cells expressing mCherry (Fig. 5f, g).

**Enrichment of base editing in primary T cells.** Genome editing is being used to explore the therapeutic potential of primary T cells, a highly clinically relevant cell type[62]. However, genome editing in T cells suffers from low efficiencies[10,11]. To determine whether the DT-HBEGF selection system could be used to facilitate the engineering of primary human T cells, we tested for co-selection of cytidine base editing events. We designed three sgRNAs that introduce premature stop codons into *PDCD1* (programmed cell death protein 1), *CTLA4*, and *IL2RA*, all of which are involved in immune regulation[11,58,63]. Each sgRNA was co-electroporated with purified CBE3 protein and synthetic sgRNA10 as an assembled ribonucleoprotein complex (RNP) into isolated CD4 + T cells. Analysis of the targeted genomic loci showed ~1.7-fold increase in base editing efficiency after DT

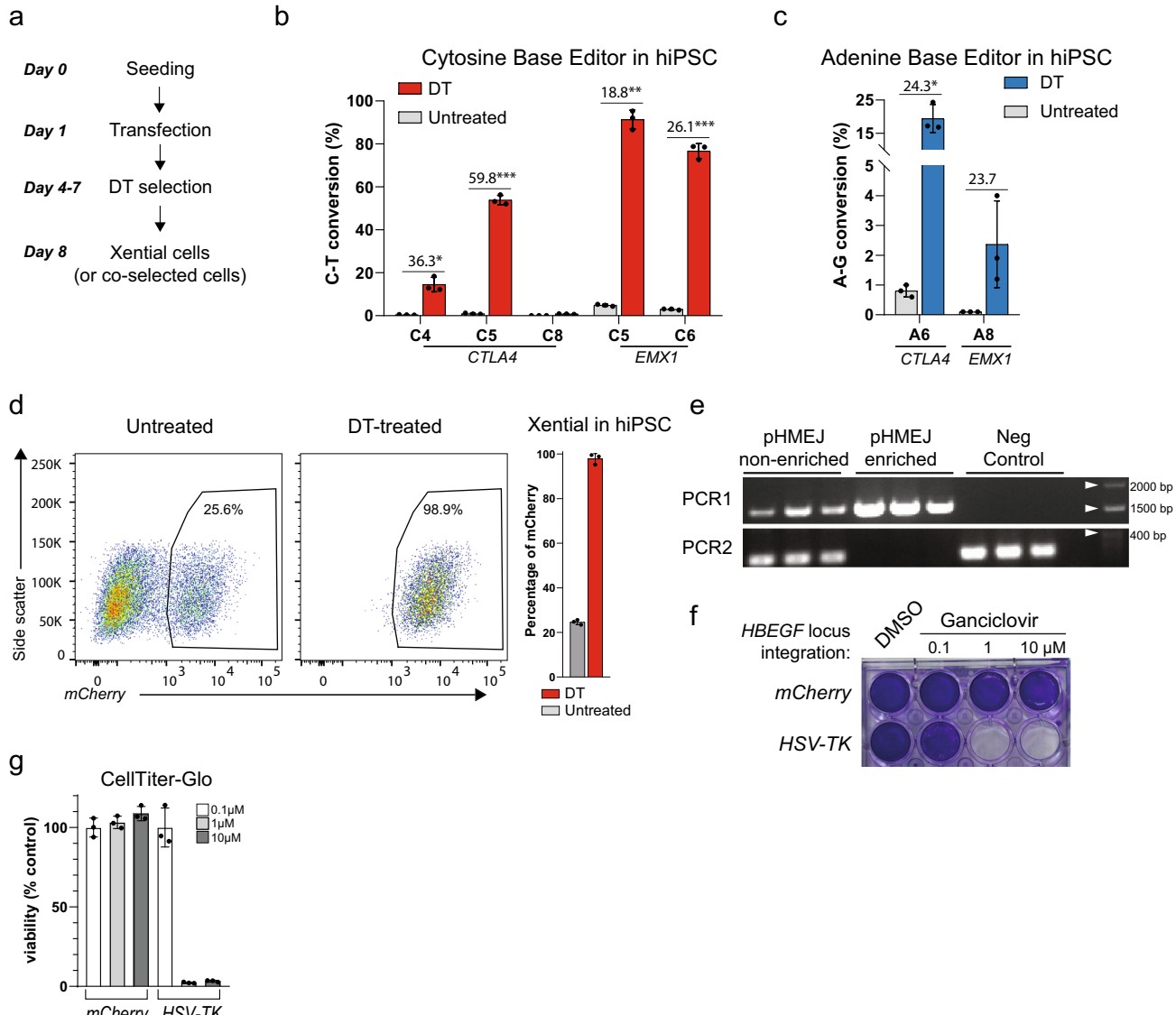

**Fig. 5 Enrichment of genome editing events in human iPSCs. a** Schema of the experiment timeline. **b, c** Co-selection of CBE (**b**) and ABE (**c**) editing events in the genome of hiPSCs cultured with or without DT. Analysis of NGS reads showing containing **b** C–T conversions or **c** A–G conversions as a percentage of full-length reads from indicated loci using depicted sgRNAs. **d** Enrichment of knock-in events at *HBEGF* locus with Xential. The left panel shows the mCherry signal in the flow cytometry scatter plots for non-enriched and DT-enriched samples, and the right panel shows the quantitative frequencies of mCherry-positive cells. **e** PCR analysis of genomic DNA from hiPSCs after Xential. Genotyping at *HBEGF* intron 3 (see Fig. 3d). Representative results were shown from three independent biological replicates. **f–g** The HSV-TK gene inserted in the *HBEGF* locus using Xential renders hiPSCs sensitive to gancivlovir. Crystal violet staining assay (**f**) and CellTiter-Glo assay (**g**) showing viability of cells containing either the mCherry or the HSV-TK gene inserted in the *HBEGF* locus. Cells were treated with depicted concentration of ganciclovir or DMSO (control) for 3 days followed by a 3-day recovery without the drug. Values and error bars reflect mean ± s.d. of $n = 3$ independent biological replicates. Relative fold changes are indicated in the graphs. *$P < 0.05$, **$P < 0.01$, ***$P < 0.001$, Student's paired $t$ test (two-tailed). $P$ values are calculated as below: in **b**, *CTLA4* (C4 = 0.0179 and C5 = 0.0007) and *EMX1* (C5 = 0.0010 and C6 = 0.0006); in **c**, *CTLA4* (0.0154) and *EMX1* (0.1146). Source data of Fig. 5b–e, g are provided as a Source data file.

selection, compared to nonselected cells (Fig. 6). Thus, the DT-HBEGF selection system works effectively in non-cancer-transformed cells, such as human iPSCs and primary T cells, suggesting it may offer a method for enriching engineered cells for therapeutic purposes.

**Enrichment of base editing events in vivo by co-selection.** Transgenic mice expressing human *HBEGF* under a tissue-specific promoter have been developed as a tool to study tissue/cell function in vivo[64], given DT ablates only the cells expressing the hHBEGF protein[64]. This transgenic mouse model enabled us

to test DT-HBEGF selection in vivo. We used DT-based co-selection with CBE3 in a humanized mouse model expressing hHBEGF under the liver cell-specific albumin promoter. As a target for genome editing, we choose the mouse *Pcsk9* gene, using a previously validated sgRNA that introduces a premature stop codon[8]. The *Pcsk9* sgRNA was delivered together with CBE3 and the sgRNA targeting human *HBEGF* using adenovirus AdV8 (Fig. 7a). Two weeks after AdV8 injection mice were treated with DT and divided into two groups. The control, non-enriched background was sacrificed at 24 h, before DT toxicity is observed[64]. The enriched group was sacrificed 4–11 days after DT treatment (Fig. 7a and Supplementary Fig. 10). The mice

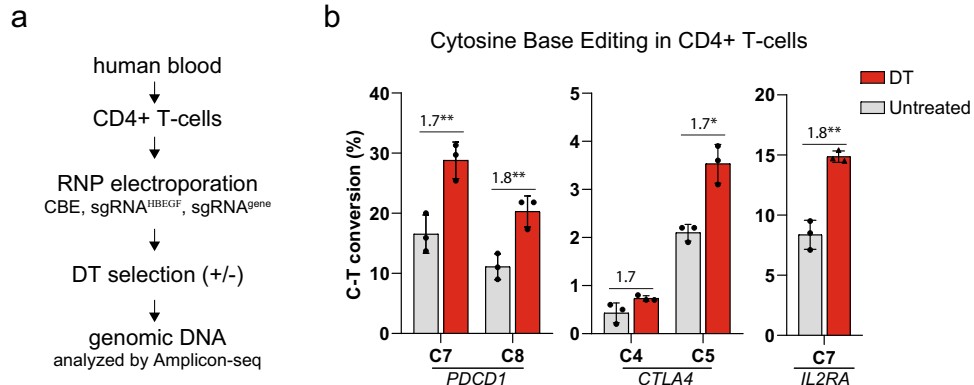

**Fig. 6 Co-selection of CBE editing events in human primary T cells using DT selection. a** Schema of the experimental setup. Total primary CD4+ T cells isolated from human blood, electroporated with CBE3 proteins, synthetic sgRNA10, and a synthetic sgRNA targeting a second genomic locus. The cells were cultivated with or without DT (untreated). **b** Analysis of NGS reads shown containing C–T conversion as a percentage of full-length reads in Amplicon-seq at indicated loci using depicted sgRNAs. The presented values and error bars show mean ± s.d. of n = 3 independent biological replicates. Relative fold changes are indicated in the graphs. *P < 0.05, **P < 0.01, ***P < 0.001, Student's paired t test (two-tailed). P values are calculated as below: in **b**, *PDCD1* (C7 = 0.0040 and C5 = 0.0075), *CTLA4* (C4 = 0.0955 and C6 = 0.0459), and *IL2RA* (C7 = 0.0071). Source data of Fig. 6b are provided as a Source data file.

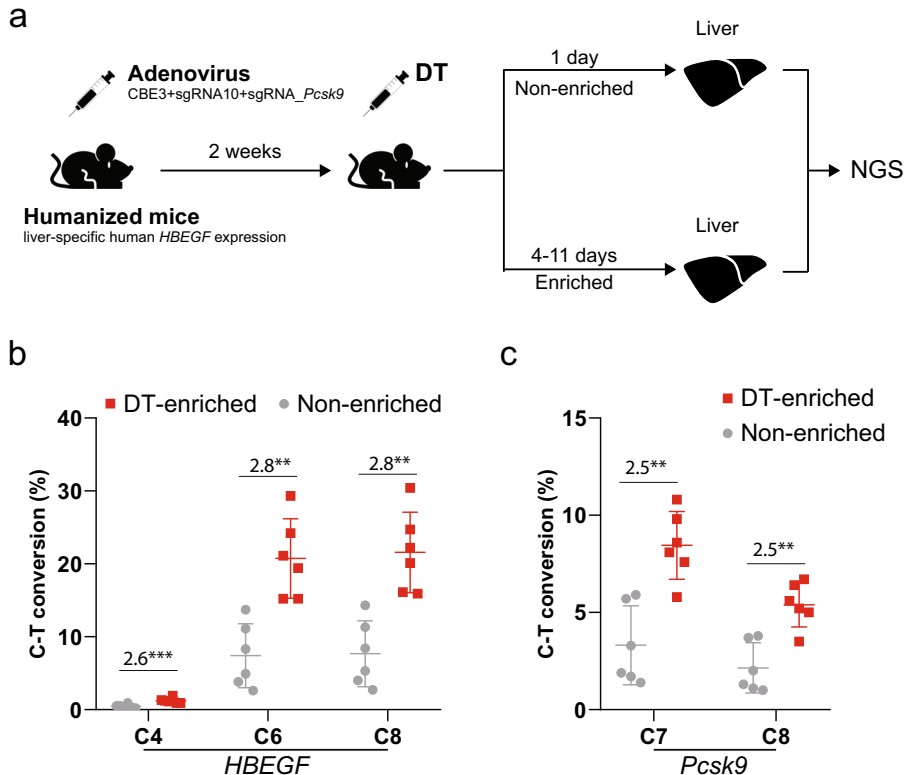

**Fig. 7 Co-selection of base editing events in vivo in mice with humanized liver. a** Design of an in vivo co-selection experiment. Adenovirus was used to introduce CBE3, sgRNA10, and a sgRNA targeting *Pcsk9*. After 2 weeks, mice were dosed with DT and terminated at 24 h or 4–11 days. At termination, liver tissue was collected for genomic DNA extraction and analysis by NGS. Icons of mice and syringes were created in a previous publication[84], and were used without modification under Creative Commons Attribution 4.0 International License (http://creativecommons.org/licenses/by/4.0/). **b** Enrichment of CBE editing at the *HBEGF* locus. **c** Co-selection of the CBE editing events at the *Pcsk9* locus. The presented values and error bars present mean ± s.d. of n = 6 independent biological replicates. The relative fold change values are indicated in the graphs. *P < 0.05, **P < 0.01, Student's paired t test (two-tailed). P values are calculated as below: in **b**, *HBEGF* (C4 = 0.0007, C6 = 0.0022, and C8 = 0.0017); in **c**, *Pcsk9* (C4 = 0.0100 and C6 = 0.0099). Source data of Fig. 7b, c are provided as a Source data file.

terminated at 11 days presented mild-to-moderate liver damage, histologically. Genomic DNA extracted from liver tissues showed 2.5–2.8- fold increase in base editing efficiency at the human

*HBEGF* transgene and at the *Pcsk9* gene, compared to control mice, indicating that co-selection of Cas9-driven genome editing events can be achieved using a bacterial toxin in vivo.

## Discussion

In this study, we leveraged the specific interaction between DT and its receptor HBEGF, and the toxin's potency in inducing cell death[30] to develop a powerful co-selection genome editing system. Our approach relies on the DNA editing of the *HBEGF* gene, by Cas9 or DNA base editors, to induce mutations that prevent the toxin–receptor interaction, such that the toxin can no longer be taken up by or kill the edited cell. *HBEGF* editing co-selects for second-site editing, presumably because edited cells are permissive to DNA manipulation at other loci.

In comparison with other selection methods, our toxin-based method offers five key advantages.

First, it provides a universal solution to enrich for a variety of editing events in cells, including single-nucleotide polymorphisms, small insertions and deletions, and precise large-fragment knock-ins. These edits can be achieved without the need for an exogenous selection marker, unlike other selection methods involving Cas9 nuclease or base editors[13,15,16].

Second, while most selection methods require the introduction of loss-of-function indels at the selection locus[14,20,24], we designed and engineered specific mutations in *HBEGF* that render it resistant to the toxin. These substitutions do not perturb *HBEGF* function and we could not find any detectible effect on cell fitness. Furthermore, we minimize the risk of unexpected DNA rearrangements by using either base editing or a precise DNA template-based knock-in strategy at the *HBEGF* locus[65–67].

Third, a variant of our selection method, Xential, relies on the insertion of a DNA fragment into the *HBEGF* locus that (1) introduces a specific mutation into *HBEGF* and (2) uses its transcription to express a gene of interest. Given the expression of a single wild-type allele of the HBEGF receptor would render cells sensitive to DT, Xential selects only for cells with the biallelic insertion of the transgene at the *HBEGF* locus, thus producing a homogenous population of cells. In addition, our experiments in human cells, including cancer and pluripotent stem cells, suggest that gene insertions at the *HBEGF* locus are potentially suitable for applications in cell therapy. The Xential selection strategy can be used for the rapid validation of genetic variants. Because many genetic variants can be inserted into the *HBEGF* locus relatively quickly, the effect of a number of mutations can be easily studied without the need of generating clonal lines. Unlike transgene selection methods based on an antibiotic-resistant gene, Xential does not introduce bacterial protein into the cell and, therefore, has a low immunogenicity potential, and results would not be confounded by other proteins expressed simultaneously for selection.

Fourth, we demonstrate that toxin-based selection for Cas9-driven genome editing events occurs efficiently in vivo, in humanized mice. We hypothesize that similar strategies can be used in the future to select edited human cells in vivo, for the purpose of facilitating the generation of xenograft models[68], or for producing large quantities of edited cells in animals[69,70].

Fifth, toxins are often large, modular biomolecules where the toxin subunit can be uncoupled from the targeting module[25]. This modularity could allow facile modulation of the toxin target, through coupling to an antibody or a cytokine specific for the desired cell type[29,71]. We anticipate that such chimeric toxins, or similarly antibody–drug conjugates[72], could be used for selection strategies such as that described here. Overall, such an extension of our method would not only expand the spectrum of targets engaged in selecting/co-selecting both in vitro and in vivo, but also provide translational applications. For example, simultaneous modification of membrane receptors, such as PD-1, CTLA4, or TCR[9,11,58], while selecting for and introducing a therapeutically relevant transgene could improve the therapeutic efficacy of the engineered cells[2,3,11]. Customized toxins designed to target these receptors could provide a direct selection method for the desired engineered cells. Overall, our methodology should be of utility to a broad range of cell and gene therapy applications, and to the generation of disease models.

## Methods

**Plasmids.** Plasmids expressing SpCas9 were constructed using a codon-optimized SpCas9 with a nuclear localization signal fused to a T2A peptide and puromycin acetyltransferase in the pVAX1 backbone. Two version of SpCas9 plasmids were constructed to drive the expression of SpCas9 under the control of the CMV (CMV-SpCas9) or EF1α promoter (EF1α-SpCas9). Plasmids expressing the CBE3 were synthesized employing the previously published sequence[34], and subcloned into the pcDNA3.1(+) vector backbone by GeneArt. Two versions of the plasmids were constructed to control CBE3 expression under CMV (CMV-CBE3) or EF1α promoter (EF1α-CBE3). ABE7.10 sequences were obtained from the original publication[35], and cloned into the pcDNA3.1(+) vector backbone. Individual sequence components were ordered from Integrated DNA Technologies and assembled using Gibson Assembly Cloning Kit (New England Biolabs). ABE7.10 plasmids were cloned either with CMV (CMV-ABE7.10) or EF1α promoter (EF1α-ABE7.10). Plasmids expressing CBE4max or ABE8e under the control of EF1α promoter (EF1α-CBE4max or EF1α-ABE8e) were synthesized by GenScript.

Plasmids expressing sgRNAs were cloned by replacing the target sequence of the template plasmid[73]. Complementary primer pairs containing the target sequence (5′-AAAC-N20-3′ and 5′-ACCG-N20-3′) were annealed (95 °C 5 min, then ramp down to 25 °C at 1 °C/min) and assembled with *AarI* digested template using T4 ligase (New England Biolabs). All primer pairs are listed in Supplementary Data 1. The plasmid expressing sgRNA targeting BFP or the plasmid expressing sgRNA targeting EGFR and CBE3 was described in our previous publication[15].

The plasmids used as DNA repair templates for the *HBEGF* or *HIST1H2BC* loci were synthesized by GenScript and modified, using Gibson Assembly Cloning Kit (New England Biolabs). Individual sequence components were ordered from Integrated DNA Technologies. Template plasmids for the *HBEGF* locus was designed to contain a splicing acceptor sequence[74], followed by the mutated sequence of the *HBEGF* exon 4 linked to the mCherry coding sequence with a self-cleaving peptide (T2A)[52]. The plasmids used for the tagging in the *HIST1H2BC* locus were designed to contain a GFP coding sequence followed by a self-cleaving peptide with the coding sequence of blasticidin deaminase. For both loci, pHMEJ and pHR were designed to contain left and right homology arms flanking the insertion sequence, while pNHEJ was designed to contain no homology arms (Supplementary Data 2). pHMEJ were designed to contain one sgRNA cutting site flanking each homology arm, while pHR did not (Supplementary Data 2). For comparing the puromycin selection with the DT selection, a self-cleavage puromycin resistant protein coding sequence was inserted between the *HBEGF* exon sequence and self-cleavage mCherry coding sequence (pHMEJ_PuroR, Supplementary Data 2). For the safety switch gene delivery, the mCherry gene was replaced with the HSV-TK gene (synthesized by GenScript, Supplementary Data 2).

dsDNA templates were prepared by PCR amplification of the plasmid pHMEJ with primers listed in Supplementary Data 1, followed by purification with MAGBIO magnetic SPRI beads. PCR amplification was performed using Phusion Flash High-Fidelity PCR Master Mix (Life Technologies). ssDNA templates were prepared using the Guide-it™ Long ssDNA Production System (Takara Bio) with primers listed in Supplementary Data 1. Final products were purified by MAGBIO magnetic SPRI beads and analyzed by Fragment Analyzer (Agilent). The oligo template used for *CD34* locus was ordered from IDT as PAGE purified oligo (Supplementary Data 1).

**Cell culture.** HEK293 (ATCC, CRL-1573), HCT116 (ATCC, CCL-247), and PC9-BFP[15] cells were maintained in Dulbecco's modified Eagle's medium (DMEM) supplemented with 10% fetal bovine serum (FBS). hiPSCs[75] were maintained in the Cellartis DEF-CS 500 Culture System (Takara Bio), according to manufacturer's instructions. All cell lines were cultured in 37 °C with 5% CO₂. Cell lines were authenticated by STR profiling and tested negative for mycoplasma.

**T-cell isolation, activation, and propagation.** Healthy donors were recruited from AZ volunteers and all samples were taken following appropriate blood collection guidelines. All blood donor volunteers signed Informed Consent form and donation was approved by AstraZeneca's Institutional review board and local Ethic committee (033-10). Peripheral blood mononuclear cells were isolated from fresh blood using Lymphoprep (STEMCELL Technologies) density gradient centrifugation and total CD4+ T cells were enriched by negative selection with the EasySep Human CD4+ T Cell Enrichment Kit (17952, STEMCELL Technologies). Enriched CD4+ T cells were then further sorted by fluorescence-activated cell sorting (FACSAria III, BD Biosciences) based on exclusion of CD8+ CD14+ CD16 + CD19+ CD25+ cell surface markers to an average purity of 98% (Supplementary Fig. 11). The following antibodies were purchased from BD Biosciences and used at concentrations as follows: CD4-PECF594 (RPA-T4, 0.5 μg/mL), CD25-PECy7 (M-A251, 8 μg/mL), CD8-APCCy7 (RPA-T8, 2 μg/mL), CD14-APCCy7

(MφP-9, 8 µg/mL), CD16-APCCy7 (3G8, 8 µg/mL), CD19-APCCy7 (SJ25-C1, 1 µg/mL), and CD45RO-BV510 (UCHL1, 1 µg/mL). Cell sorting was performed using a FACSAria III (BD Biosciences).

The CD4+ T cells were propagated in RPMI-1640 medium containing the following supplements: 1% (v/v) GlutaMAX-I, 1% (v/v) nonessential amino acids, 1 mM sodium pyruvate, 1% (v/v) L-glutamine, 50 U/mL penicillin and streptomycin, and 10% heat-inactivated FBS (all from Gibco, life Technologies). The T cells were activated using the T Cell Activation/Expansion kit (130-091-441, Miltenyi). To this end, $1 \times 10^6$ cells/mL were activated at bead to cell ratio of 1:2 and $2 \times 10^5$ cells per well were seeded into round-bottom tissue culture treated 96-well plates for 24 h. Cells were pooled prior to electroporation.

**Cell transfections.** Twenty hours prior transfections $1.25 \times 10^5$ or $6.75 \times 10^4$ HEK293, HCT116, and PC9-BFP cells were seeded into 24-well or 48-well plates, respectively. Transfections were performed with FuGENE HD transfection reagent (Promega) using a 3:1 transfection reagent to plasmid DNA ratio. For 24-well plate formats, the amount and weight ratios of transfected DNA are listed in Supplementary Table 2 and Supplementary Table 3. For 48-well plate formats, the amount of DNA was reduced by half.

The iPSCs cells were transfected with FuGENE HD using a 2.5:1 transfection reagent to DNA ratio and a reverse transfection protocol. For transfections, $4.2 \times 10^4$ cells were seeded per well in 48-well format directly onto prepared transfection complexes as described in Supplementary Table 4.

The CD4+ T cells were electroporated with RNPs using the 10 µL Neon transfection kit (MPK1096, Thermo Fisher). CBE3 proteins were produced using the method described before[76]. An extra purification step was performed on a HiLoad 26/600 Superdex 200 pg column (GE Healthcare) with a mobile phase consisting of 20 mM TrisCl pH 8.0, 200 mM NaCl, 10% glycerol, and 1 mM TCEP. Purified CBE3 protein was concentrated to 5 mg/mL in a Vivaspin protein concentrator spin columns (28932363, GE Healthcare) at 4 °C, before flash freezing in small aliquots in liquid nitrogen. RNPs were prepared as follows; 20 µg CBE3 protein, 2 µg of target sgRNA and 2 µg of selection sgRNA (TrueGuide Synthetic gRNA, Life Technologies), and 2.4 µg electroportation enhancer oligos (HPLC-purified, Sigma; Supplementary Data 1) were mixed and incubated for 15 min. Cells were washed with PBS and resuspended in buffer R at a concentration of $5 \times 10^7$/mL. A total of $5 \times 10^5$ cells were electroporated with RNPs using the following settings: voltage: 1600 V, width: 10 ms, and pulse number: 3. After electroporation cells were incubated over night in 1 mL of RPMI medium complemented with 10% heat-inactivated FBS in a 24-well plate. The next day cells were collected, centrifuged at $300 \times g$ for 5 min, resuspended in 1 mL of complete growth medium containing 500 U/mL IL-2 (PHC0026, Prepotech) and split into five wells of a round-bottom 96-well plate.

**Diphtheria toxin treatments in vitro.** Transfected HEK293, HCT116, and PC9-BFP cells were selected with 20 ng/mL DT at days 3 and 5 after transfections. iPSCs were treated with 20 ng/mL DT from day 3 after transfections. DT-supplemented growth medium was exchanged daily until negative control cells died. Transfected CD4+ T cells, were treated with 1000 ng/mL DT at days 1, 4, and 7 after electroporation.

**Recombinant HBEGF purification from bacteria.** The *HBEGF* gene fragment encoding a soluble human wild-type HBEGF (amino acids 63–119) or the E141K mutant were cloned to pET32a (Novagen) and expressed in BL21(DE3) strain. Briefly, the recombinant proteins were induced by adding 0.4 mM IPTG for 24 h at OD600 0.6. The collected cells were lysed in lysis buffer (20 mM Tris–HCl, 500 mM NaCl, and 1% Triton X-100) and sonicated. Both proteins containing Trx-6xHis tags were purified over Ni-NTA column and elute with imidazole (PBS, pH 7.5, 500 mM imidazole)[77,78]. The precleared lysate, the SDS–Page gel electrophoresis followed by the Coomasie staining was used to assess the purity of the protein purification.

**Recombinant HBEGF activity assay.** HEK293 cells were plated in six-well plate ($1.5 \times 10^6$) and grown for 24 h. The attached cells were washed three times with PBS and cultured for 12 h in DMEM media without FBS (serum starvation). Subsequently, DMEM containing the recombinant HBEGF (10 ng/mL) or the elution buffer used for purification was added to cells followed by 5 min incubation at 37 °C. The cells were lysed (25 mM Hepes pH 7.4, 150 mM NaCl, 1%Triton, protease inhibitors (Roche), and phosphatase inhibitors (Roche)) on ice and protein was extracted for SDS–PAGE followed by western blot[79].

**Cell viability and proliferation assays.** Cell viability was analyzed using the AlamarBlue cell viability reagent (Thermo Fisher) or CellTiter-Glow (Promega) according to the manual. Fluorescence emission was recorded with a SpectraMax iD3 Multi-Mode Microplate Reader (Molecular Devices). To determine whether introduced mutations into the *HBEGF* locus affect the proliferative capacity, we evaluated cell growth of HEK293 wild-type cells and the produced *HBEGF*-mutant sublines. To monitor proliferation curves, 2000 cells were seeded per well of a 96-well plate and cell confluence was recorded every 24 h for 7 days, using the Incucyte S3 live-cell analysis system (Essen BioScience). For the experiments with the HSV-

TK safety switch in hiPSCs, ganciclovir (Sigma, SML2346) was included in the cell culture media at 0.1, 1, and 10 µM concentration for 3 days, followed by a 3-day recovery.

**PCR analysis.** PCR analysis was performed to discriminate between successful knock-in into *HBEGF* intron 3 (PCR1) and the wild-type sequence (PCR2). PCR reactions were carried out in 20 µL volume using 1.5 µL of extracted genomic DNA as template. Phusion Flash High-Fidelity PCR Master Mix (Thermo Fisher) and the recommended protocol was applied with a final primer concentrations of 0.5 µM. Primer pair PCR1_fwd and PCR1_rev was used for PCR1 to detect knock-in junctions (annealing temp: 62 °C, elongation time: 1 min), and primer pair PCR2_fwd and PCR2_rev was used for PCR2 to detect the wild-type *HBEGF* intron (annealing temp: 64.5 °C, elongation time: 5 s). Sequences of primer pairs are provided in Supplementary Data 1. For PCR2, the elongation time was set to 5 s to favor amplification of the wild-type *HBEGF* intron 3 PCR product (280 bp) over the integrant PCR product (2229 bp). PCR products were analyzed through agarose gel electrophoresis. For further analysis of the junction between inserted DNA and genomic DNA in Xential engineered cells, PCR was performed using the Left_F/Left_R primer pair (forward insertion, PCR_L) or the Left_F/Right_F primer pair (reverse insertion, PCR_Lr). Conditions for the PCR reactions and sequences of primers are provided in Supplementary Data 1.

**Flow cytometry analysis.** The frequency of cells expressing mCherry and GFP was assessed with a BD Fortessa (BD Biosciences), and flow cytometry data were analyzed with the FlowJo software (Three Star).

**Genomic DNA extractions and next-generation Amplicon sequencing.** Genomic DNA was extracted from cells 3 days after transfections or after completed DT selection using QuickExtract DNA extraction solution (Lucigen), according to the manual. Amplicons of interest were analyzed from genomic DNA samples on a NextSeq platform (Illumina). In brief, genomic sites of interest were amplified in a first round of PCR using primers that contained NGS forward and reverse adapters (Supplementary Data 1). The first PCR was setup using NEBNext Q5 Hot Start HiFi PCR Master Mix (New England Biolabs) in 15 µL reactions, with 0.5 µM of primers and 1.5 µL of genomic DNA as template. PCR was carried out applying the following cycling conditions: 98 °C for 2 min, 5 cycles of [98 °C for 10 s, annealing temperature for each pair of primers for 20 s (calculated for genomic binding regions of primers by NEB Tm Calculator), and 65 °C for 10 s], then 25 cycles of [98 °C for 10 s, 98 °C for 20 s, and 65 °C for 10 s], followed by a final 65 °C extension for 5 min. PCR products were purified using HighPre PCR Clean-up System (MagBio Genomics) and correct PCR product size, and DNA concentration was analyzed on a Fragment Analyzer (Agilent). Unique Illumina indexes were added to PCR products in a second round of PCR using KAPA HiFi Hotstart Ready Mix (Roche). Indexing primers were added in a second PCR step and 1 ng of purified PCR product from the first PCR was used as template in a 50 µL reaction volume. PCR was performed applying the following cycling conditions: 72 °C for 3 min, 98 °C for 30 s, then ten cycles of [98 °C for 10 s, 63 °C for 30 s, and 72 °C for 3 min], followed by a final 72 °C extension for 5 min. Final PCR products were purified using HighPre PCR Clean-up System (MagBio Genomics) and analyzed by Fragment analyzer (Agilent). Libraries were quantified using Qubit 4 Fluorometer (Life Technologies), pooled and sequenced on a NextSeq instrument (Illumina).

**Bioinformatics.** NGS sequencing data were demultiplexed using bcl2fastq software, and individual FASTQ files were analyzed using a Perl implementation of the Matlab script described previously[34]. For the quantification of indels or base editing frequencies, sequencing reads were scanned for matches to two 10 bp sequences that flank both sides of an intervening window, in which indels or base edits might occur. If no matches were located (allowing maximum 1 bp mismatch on each side), the read was excluded from the analysis. If the length of the intervening window was longer or shorter than the reference sequence, the sequencing read was classified as an insertion or deletion, respectively. The frequency of insertions or deletions was calculated as the percentage of reads classified, as insertion or deletion within the total analyzed reads. If the length of this intervening window exactly matched the reference sequence the read was classified as not containing an indel. For these reads, the frequencies of each base at each locus was calculated in the intervening window and was used as the frequencies of base edits. For off-target analysis, a list of in silico predicted candidate sites was generated for sgRNA5, sgRNA10, or sgRNAin3 using Cas-OFFinder[80], respectively (Supplementary Data 3). Top three candidates were selected for each sgRNA for NGS analysis. Sequencing data were analyzed by CRISPResso2 (ref. [81]).

**Cytidine base editing and DT treatment of mice humanized for hHBEGF expression.** All mouse experiments were approved by the AstraZeneca internal committee for animal studies and the Gothenburg Ethics Committee for Experimental Animals (license number: 162–2015+) compliant with EU directives on the protection of animals used for scientific purposes. Experimental mice were generated as double heterozygotes by breeding Alb-Cre mice (016833, The Jackson

Laboratory) to iDTR mice (Expression of transgene, human *HBEGF*, is blocked by loxP-flanked STOP sequence) on the C57BL/6NCrl genetic background. Mice were housed in negative pressure IVC caging, in a temperature controlled room (21 °C) with a 12:12 h light–dark cycle (dawn: 5:30 a.m., lights on: 6:00 a.m., dusk: 5:30 p.m., lights off: 6 p.m.) and with controlled humidity (45–55%). Mice had access to a normal chow diet (R36, Lactamin AB, Stockholm, Sweden) and water ad libitum.

For base editing, 6-month-old mice, six male, and six female, were randomized into two groups with equal male and female mice in each group. Adenoviral vectors expressing CBE3, sgRNA10, and sgRNA targeting mouse *Pcsk9* ($1 \times 10^9$ IFU particles per mouse) were intravenously injected. Two weeks after virus administration, all mice received DT (200 ng/kg) intraperitoneally. Control mice were terminated 24 h after DT injection. Experimental mice were terminated 11 days after DT injection. Four mice were terminated prior to experimental endpoint as the humane endpoint of the ethics license was reached. At necropsy, liver tissues were collected for morphological and molecular analyses.

**Histology analysis**. At necropsy, mice were euthanized under isoflurane anesthesia and liver collected in 4% neutral-buffered formalin for assessment. Tissue was embedded in paraffin and prepared as 5 μm thick sections. Sections were stained for hematoxylin and eosin for morphological characterization. All histological slides were blinded and examined using light microscopy (Carl Zeiss Microscopy GmbH, Jena, Germany) by an experienced board-certified pathologist. Severity grades (0–5) were assigned according to standard grading criteria as per Schafer et al.[82] with 0 = lesion not detected, 1 = minimal, 2 = mild, 3 = moderate, 4 = marked, and 5 = severe.

**Translocation analysis**. HEK293 cells were transfected with different combinations of sgRNAin3, sgRNA targeting *HIST1H2BC* and pHMEJ repair templates described in Supplementary Table 5. Genomic DNA was isolated from these cells with or without DT selection with Gentra Puregene Cell Kit (Qiagen) and was diluted to 10 ng/μL for ddPCR analysis. A FAM-labeled ddPCR assay was designed for detecting the the balanced translocation between *HBEGF* and *HIST1H2BC*, using Primer 3 Plus[83] and was ordered as custom assay from BioRAD (sequence information in Supplementary Table 6). A Mastermix was prepared using a final concentration of 1× ddPCR Supermix for Probes, no dUPT (186-3024, Bio-Rad), 1× FAM-labeled HBEGF-HIST1H2BC assay (custom assay 10031276, BioRAD), 1× AP3B1-HEX labeled human reference assay (dHsaCP1000001, BioRAD), and 1/40 HaeIII (15205016, Invitrogen). A total of 20 μL Mastermix per well to be analyzed was prepared in ultrapure RNase and DNase free water (10977–035, Invitrogen) with 5 μL 10 ng/μL genomic DNA. An automated Droplet Generator (BioRAD) was used to generate droplets in a new semi-skirted 96-well PCR plate (30129504, Eppendorf). After droplet generation, the PCR plate was placed in a C1000 Touch™ Thermal Cycler (Bio-Rad, cat no. 185-1197) for PCR amplification, as detailed in Supplementary Table 6. The droplet reading was performed with the QX 100 Droplet reader (Bio-Rad, cat. no. 186-3001), using ddPCR™ Droplet Reader Oil (Bio-Rad, cat. no. 186-3004). Data acquisition and analysis was performed using the software QuantaSoft (Bio-Rad) and the "RED" program. The fluorescence amplitude threshold was set manually as the midpoint between the average fluorescence amplitude of the four droplet clusters (Translocation-positive, AP3B1-positive, positive for both targets, and empty droplets). The same threshold was applied to all the wells of the ddPCR plate.

**Trilineage differentiation assay**. Differentiation potential of hiPSCs and *HBEGF*-mutant pools into the three germ layers was assayed with the STEMdiff Trilineage Differentiation Kit (STEMCELL Technologies). In brief, cells were plated onto Cellartis DEF-CS 500 COAT-1 (Cellartis) coated six-well plates, and treated with endoderm or mesoderm differentiation media for 5 days or ectoderm differentiation media for 7 days.

RNA was extracted with the RNeasy® Plus Mini Kit (Qiagen) according to the manual. cDNA synthesis was conducted with the High Capacity cDNA Reverse Transcription Kit (Thermo Fisher) in 20 μL reactions in the presence of 40 U RNaseOUT Recombinant Ribonuclease Inhibitor (Invitrogen) following the guidelines. Gene expression analysis of pluripotency- and lineage-associated genes was performed with the following Taqman assays; ACTB (Hs01060665_g1), GAPDH (Hs02758991_g1), POU5F1 (Hs00999632_g1), NANOG (Hs02387400_g1), T (Brachyury) (Hs00610080_m1), HAND1 (Hs00231848_m1), SOX17 (Hs00751752_s1), FOXA2 (Hs00232764_m1), PAX6 (Hs01088114_m1), and OTX2 (Hs00222238_m1; all from Thermo Fisher). qPCR was carried out in white 384-well plates (Thermo Fisher) in 10 μL reactions using the 2× TaqMan® Fast Advanced Master Mix (Thermo Fisher) and 12 ng of cDNA template, applying the following thermal protocol: 95 °C for 20 s: 40 cycles of: 95 °C for 1 s, 60 °C for 20 s (QuantStudio™ 7 Flex Real-Time PCR System, Thermo Fisher). Quantification cycles (Cq's) were determined via the threshold method (QuantStudio™ Real-Time PCR Software v1.3, Thermo Fisher). Gene expression data were analyzed with the GenEx v7.1 software (MultiD Analyses AB).

**Reporting summary**. Further information on research design is available in the Nature Research Reporting Summary linked to this article.

## Data availability
Data supporting the findings of this study are presented within the article and supplementary figures. NGS data are available in the NCBI Sequence Read Archive database (BioProject accession code PRJNA684443). Additional details and data to support the findings of this study are available from the corresponding authors upon reasonable request. Source data are provided with this paper.

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

## Acknowledgements

We thank Barry Rosen and Steve Rees for scientific and managerial support to the project, John Wiseman for the help with mouse work, Pei-Pei Hsieh and Elke Ericson for the help with ddPCR, Elke Ericson and Cristina Rocha for providing guidance with

Diptheria toxin, Philippe Soriano for providing the sequence of adenovirus splicing acceptor, and Guy Riddihough (Life Science Editors) for help with editing the manuscript. This project has received funding from the European Union's Horizon 2020 research and innovation program under the Marie Skłodowska-Curie grant agreement no. 765269 (S.W) and 814316 (B.L.).

## Author contributions

S.L. and M.M. developed the original idea. S.L., N.A., and S.C. performed most of the experimental work with help from M.P., S.W., A.L., C.M., G.C., and G.S. M.F. performed bioinformatic analyses. E.G. purified proteins. G.P. and M.S. performed histology analyses. N.A., B.L., and A.S. performed hiPSC differentiation. X.X. generated the mice model. S.L. and G.S. prepared the manuscript with input from N.A., S.C., M.P., and M.M. L.S.P., M.A.C., S.M., M.B.-Y., and B.J.T. helped with the study design. M.M. supervised the study.

## Competing interests

S.L., N.A., S.C., M.P,. S.W., A.L., C.M., M.F., E.G., B.L., A.S., L.S.P., M.A.C., G.P., M.S., M.B.-Y., B.J.T., G.S., and M.M. are employees and shareholders of AstraZeneca. M.M. and S.L. are listed as co-inventors in a Astrazeneca patent application (application number: EP2020060250W) related to this work. The other authors claim no competing interests.
