## [Peer Review File · Nature Communications]

Reviewers' Comments:

Reviewer #1:

Remarks to the Author:

This manuscript describes methods to target the DT receptor gene for base editing and knock-ins. Targeted cells can be enriched by their resistance to DT, to which the receptor can no longer bind. They succeeded in achieving bi-allelic targeting with nearly 100% efficiency. They further applied DT targeting for the enrichment of targeting at a second gene of interest. The methods work in multiple cell lines, human stem cells, and humanized mouse.

The quality of data is in general good and convincing. There is only one major suggestion: for knock-in experiments, some sequencing data should be shown to further confirm that the products are consistent with homologous recombinational repair. In particular, they are necessary to determine the accuracy of repair. Conceivably, mistakes might be introduced into the products after repair, which are not necessarily reflected in cell sorting data.

Reviewer #2:

Remarks to the Author:

In "Universal toxin-based selection for precise genome editing in human cells", Li et al., described the development of a universal selection strategy, based on the introduction of DT-resistant mutations into HBEGF gene and then, selection for the edited cells with DT, therefore leading to the depletion of cells that don't have the desired genome modification. Overall, the strategy is feasible, the paper is well written and logically presented. However, there are a few points that should be addressed to warrant publication in Nature Communications.

1 – Upon T cells transfection and selection with DT, the number of cells that survive is not clear in the manuscript. You can have about 100% of desired editing upon selection, but would you have enough cells for downstream applications?

2 – Multiplex editing in T cells is highly desired in many therapeutic applications and many studies rely on knock-in into the TRAC locus and knock-out of T cells exhaustion genes for example. Previous publications demonstrated genomic rearrangements, such as translocations, as an outcome of multiple editing in T cells (Stadmauer, E. et al. CRISPR-engineered T cells in patients with refractory cancer. *Science*, 2020). This could be experimentally addressed by the authors in the session "Enrichment of knock-out and knock-in by Xential", to provide a sense of whether targeting the HBEGF gene would be prone to translocations. Additionally, as the authors propose the ideal target sites, it would be useful to know ahead what are the potential off-target sites for sgRNAIn3, sgRNA5, sgRNA10.

3 – Did the authors detect additional editing outcomes in the knock-in experiments (especially when using DNA oligos as repair template)?

4 – It could be premature to call HBEGF a safe harbor locus, as other genomic requirements need to be addressed to that end (Sadelain M, Papapetrou E, & Bushman F. Safe harbours for the integration of new DNA in the human genome. *Nat Rev Cancer*, 2012).

5 - In gauging the significance of this work, it would be valuable to see the expression of a transgene of therapeutic interest in addition to the fluorescent proteins. Interesting options to demonstrate the feasibility of the system in an extremely relevant therapeutic context would be to delivery a correct copy of HBB in HSCs or knock in a CAR in T cells.

6 - The authors mention that one of the bottlenecks of the genome editing field is the low editing rates in some relevant cells, as T cells, especially using base editors. The authors clearly showed

that they were able to increase those numbers upon selection (although modestly, 1.7 and 1.8-fold). However, new versions of base editors that perform better than their ancestors were described recently. Based on this, would the DT-selection system still outperform the new versions of ABE and CBE?

7 - Do the authors predict that the results of the HBEGF activity assay would apply for primary cells as well, leading the conclusion that the mutations in the HBEGF do not perturb gene function and that there's no effect on cell fitness?

We thank all the reviewers for the thoughtful and insightful comments on our manuscript. We have made a sincere effort to address each of these comments. These remarks have prompted several new experiments, which considerably advance and extend our manuscript. This document contains a detailed point-by-point response to the reviews.

Reviewer #1 (Remarks to the Author):

This manuscript describes methods to target the DT receptor gene for base editing and knock-ins. Targeted cells can be enriched by their resistance to DT, to which the receptor can no longer bind. They succeeded in achieving bi-allelic targeting with nearly 100% efficiency. They further applied DT targeting for the enrichment of targeting at a second gene of interest. The methods work in multiple cell lines, human stem cells, and humanized mouse.

We would like to thank Reviewer 1 for reading and reviewing our study.

The quality of data is in general good and convincing. There is only one major suggestion: for knock-in experiments, some sequencing data should be shown to further confirm that the products are consistent with homologous recombinational repair. In particular, they are necessary to determine the accuracy of repair. Conceivably, mistakes might be introduced into the products after repair, which are not necessarily reflected in cell sorting data.

We have observed that the two most optimal templates for bi-allelic insertion are pHMEJ and pHR (Fig. 3b). Using these DNA templates, we obtain an efficient knock-in of the *mCherry* gene into the *HBEGF* locus (5% for pHR and 35% for pHMEJ positive cells before DT selection and nearly 100% of the mCherry-positive cells after selection).

In this revision, we further investigated the fidelity of the DNA insertions after applying the Xential selection. We focused on the left arm of the DNA cassette that contains the mutated Exon4 and Exon5 of the *HBEGF* gene (Fig. R1a). We first performed PCR analysis on the left junctions between the genomic DNA and the inserted DNA, and observed a PCR product that matches the expected molecular size of the DNA inserted in the forward orientation, but no product was detected once we examined insertion in the reverse orientation (Fig. R1b). This suggests an unidirectional insertion profile regardless of the DNA templated used (Fig. R1a).

We performed Amplicon-seq to further analyze the underlying DNA sequence at these junction (Fig. R1b). For both pHMEJ and pHR, the dominant sequence represents a precise insertion (95% of reads, Fig. R1c). The frequencies of other variants are close to or below the sequencing error of NGS¹.

It was reported before that a small fraction of HMEJ-mediated DNA insertion events occurs via the NHEJ pathway². In our PCR genotyping analysis, we observed an addition, faint band of a higher molecular size in the pHMEJ sample only (Fig. R1c). Amplicon-seq analysis of its

sequence identified a dominant variant containing an additional T insertion, which suggests the NHEJ-mediated DNA repair. The data was added to Supplementary Figure 5.

Altogether, for precise DNA insertions in cells proficient for the HR pathway, we advise to use the pHR template. Otherwise, the pHMEJ template serves as an attractive alternative, especially given that the already high precision of the HMEJ-mediated insertions can be further improved by either (a) optimizing the template design and (b) applying the compounds modulating DNA repair³.

Figure R1. Analysis of the DNA sequence at the junction between inserted DNA and genomic DNA at the *HBEGF* locus. (a) PCR primers were designed to amplify all types of left junctions of the insertion formed by the HR or NHEJ pathways. DT-selected samples with the pHMEJ- or pHR-mediated Xential insertions were analyzed; wild-type cells served as a negative control. (b) DNA electrophoresis on agarose gel presenting genotyping results. PCR products matching the size of HR junctions are marked by an arrow. PCR products of different sizes are indicated by a red rectangle. (c) Amplicon-seq results presenting DNA sequences around the left HR junctions

after the pHR or pHMEJ insertions. The top three variants are shown with their corresponding frequencies. Insertions, deletions or substitutions are highlighted in red. (c) Amplicon-seq results presenting the DNA sequences around the left NHEJ junctions were amplified and analyzed by NGS. Top three variants are shown with their corresponding frequencies. Insertions, deletions or substitutions are highlighted in red.

Reviewer #2 (Remarks to the Author):

In “Universal toxin-based selection for precise genome editing in human cells”, Li et al., described the development of a universal selection strategy, based on the introduction of DT-resistant mutations into HBEGF gene and then, selection for the edited cells with DT, therefore leading to the depletion of cells that don't have the desired genome modification. Overall, the

strategy is feasible, the paper is well written and logically presented. However, there are a few points that should be addressed to warrant publication in Nature Communications.

We thank Reviewer 2 for the encouraging comment and recognition of our findings presented in the manuscript.

1 – Upon T cells transfection and selection with DT, the number of cells that survive is not clear in the manuscript. You can have about 100% of desired editing upon selection, but would you have enough cells for downstream applications?

After electroporation of the primary T-cells (0.5 million) with CBE3 and sgRNA16, we obtained 11% of desired edits at the *HBEGF* locus. This generates a population of cells containing the resistant mutations that enter the DT selection step. After finishing the DT selection procedure (day 8, after 3 rounds of DT selection), the cell number reached ~75% of the untreated condition (representative flow cytometry analysis of a small fraction of DT-treated and untreated cells) (Fig. R2). While the DT selection removes the unedited cells - hence has an impact on cell number, but only minor – the quantity and viability data of the surviving cells suggests that they recovered and continue to proliferate. Therefore, in the case when a large number of cells is required for a downstream application, we envision a straightforward, scaled-up experiment to achieve even greater number of edited cells. This number can be further increased by more efficient base editors recently described^{4,5}.

Figure R2. The number and viability of the CD4+ T-cells. The primary T-cells described in the co-selection experiment with CBE (Fig. 6) were analyzed by flow cytometry at the day 8 after electroporation. (a) The left panel shows the number of viable lymphocytes recorded in the analysis of 30000 events. (b) The right panel shows the viability of the gated lymphocytes. Values and error bars reflect mean \pm s.d. of $n=3$ independent biological replicates.

2 – Multiplex editing in T cells is highly desired in many therapeutic applications and many studies rely on knock-in into the TRAC locus and knock-out of T cells exhaustion genes for example. Previous publications demonstrated genomic rearrangements, such as translocations, as an outcome of multiple editing in T cells (Stadmauer, E. et al. CRISPR-engineered T cells in patients with refractory cancer. *Science*, 2020). This could be experimentally addressed by the authors in the session “Enrichment of knock-out and knock-in by Xential”, to provide a sense of whether targeting the HBEGF gene would be prone to translocations.

We thank reviewer for the suggestion. To probe for translocation events occurring during genome modification at two loci simultaneously, we designed a panel of droplet digital PCR (ddPCR) assays. These detect the monocentric translocations from the *HIST1H2BC* locus to the *HBEGF* locus.

We initially analyzed samples without DT selection and observed translocations frequency of 0.038%-0.089%, when both DNA loci were targeted and cleaved by *SpCas9* (Fig. R3, left). When we included in our experiments DNA templates with homology arms provided for the *HBEGF* locus (Xential templates) and *HIST1H2BC*, we observed a reduction of these translocation events, suggesting that the presence of HDR Donors prevents translocations.

The DT selection following genomic editing and without DNA templates resulted in cell pools with increased translocations (Fig. R3, right). However, the addition of DNA templates for genomic editing (Xential-mediated co-selection) is sufficient to vastly reduce the translocation frequency (to 0.041%) to the level observed in the samples without DT selection (Fig. R3, right). This observation indicates that the Xential-mediated co-selection provides advantage over previous co-selection methods. Therefore, we advise to apply the Xential-mediated co-selection for engineering precise genome modifications with reduced risk of genomic rearrangements. The data was added to Supplementary Figure 6c.

Figure R3. Analysis of translocation events in multiplex genome editing experiment. The graph presenting the translocation frequency based on the BioRad ddPCR Copy Number Assay. ddPCR primers and probes were designed to detect the balanced translocation between *HBEGF* and *HIST1H2BC*. *AP3B1* was used as the reference assay for calculating frequencies. Values and error bars reflect mean \pm s.d. of n=3 independent biological replicates. *P< 0.05, **P< 0.01, ***P<0.001, Student's paired t-test.

Additionally, as the authors propose the ideal target sites, it would be useful to know ahead what are the potential off-target sites for sgRNA13, sgRNA5, sgRNA10.

In order to evaluate off-target profiles for the sgRNAs used in this study, we first performed *in silico* analysis with Cas-OFFinder to predict off-target sites⁶ (Table. R1). We selected top 3 predicted off-target sites for each sgRNA (sgRNA13, sgRNA5, sgRNA10) and experimentally inspected their editing outputs (ABE, CBE and Cas9 nuclease) in HEK293 and hiPSC by NGS (Fig. R4a). While the on-target guide editing frequencies were reaching ~100%, we observed no off-target editing at the sequenced sites within the limit of detection (0.1%) in both HEK293 (Fig. R4b, Supplementary Figure 3) and hiPSC cells (Fig. R4b, Supplementary Figure 8). We concluded that the sgRNAs used in the study are efficient and specific.

a

	Name	Target sequence	Off-target sequence	chromosome
ABE	sgRNA5_chr2_off	GCAAATATGTGAAGGAGCTCNGG	GCAAATATGTGAAaGA-CTCTGG	chr2
	sgRNA5_chr7_off	GCAAATATGTGAAGGAGCTCNGG	GaAAATAT-TGAAGGAGCTCTGG	chr7
	sgRNA5_chr14_off	GCAAATATGTGAAGGAGCTCNGG	GCAAATAaGTGAAGGAGC-CAGG	chr14
CBE	sgRNA10_chr1_off	CAC-CTCTCTCCATGGTAACCNGG	CAC ^A T ^A CTCTCTCCA ^g GGTAACCAGG	chr1
	sgRNA10_chr3_off	CACCTCTCTCCATGGTAACCNGG	CA-CTCTCT ^g CATGGTAACCAGG	chr3
	sgRNA10_chr10_off	CACCTCTCTCCATGGTAACCNGG	CA-CTCTCTC ^c TGGTAACCAGG	chr10
Cas9	sgRNAin3_chr4_off	GGGTGATGTTGCCTGACCGGNGG	G-cTGATGTTGCCT ^a ACCGGGGG	chr4
	sgRNAin3_chr5_off	GGGTGATGTTGCCTGACCGGNGG	GGGTGAT-TTGCTGA ^{at} GGAGG	chr5
	sgRNAin3_chr12_off	GGGTGATGTTGCCTGACCGGNGG	GGGTG ^g TGTTGCCTG-CC ⁱ GTGG	chr12

b

Fig. R4. The Off-target analysis of sgRNA5, sgRNA10 and sgRNAin3. (a) Table presenting top 3 off-target sites selected for analysis. Mismatches or DNA/RNA bulges are highlighted in red. (b) Data presenting efficiency of ABE, CBE and Cas9 at depicted loci analyzed by NGS. Non-targeting sgRNA does not target human genome and was used as negative control in this experiment. For sgRNAin3, the percentage of the mCherry knock-in cells is presented as an estimate of the sgRNA on-target efficiency (to the right). Values and error bars reflect mean \pm s.d. of n=3 independent biological replicates.

3 – Did the authors detect additional editing outcomes in the knock-in experiments (especially when using DNA oligos as repair template)?

In order to address this question, we analyzed the NGS data of this experiment. DT treatment enriches the insertion of the DNA oligo. In addition, we detected several editing outcomes in the DNA oligo knock-in experiment at the *CD34* locus (Fig. S4c) that overall qualitatively and quantitatively differ in between the untreated and DT-treated cells. Below, we summarize the analysis of the DNA sequencing using RIMA⁷ (Fig. R5).

Figure R5. The DNA sequence variants in the oligo knock-in experiment at the *CD34* locus. Sequencing data of the top 20 variants are shown for each sample (untreated left and DT-treated right) with their total frequency.

4 – It could be premature to call HBEGF a safe harbor locus, as other genomic requirements need to be addressed to that end (Sadelain M, Papapetrou E, & Bushman F. Safe harbours for the integration of new DNA in the human genome. Nat Rev Cancer, 2012).

We appreciate the comment of the reviewer and agree on this point. We removed the statement about the safe harbor locus from the text.

5 - In gauging the significance of this work, it would be valuable to see the expression of a transgene of therapeutic interest in addition to the fluorescent proteins. Interesting options to demonstrate the feasibility of the system in an extremely relevant therapeutic context would be to delivery a correct copy of HBB in HSCs or knock in a CAR in T cells.

We would like to thank the reviewer. To demonstrate the translational potential of our method, we chose to install a safety switch, as a transgene of therapeutic interest⁸. Our rationale was that the transgene will be stably expressed in this locus, because of (a) the bi-allelic insertion of the DNA cassette and (b) the importance of the *HBEGF* gene for cell growth.

To this end, we generated a plasmid (DNA template donor for the DNA repair) containing gene encoding for Thymidine Kinase from Herpes Simplex Virus (HSV-TK). We applied Xential to introduce this HSV-TK transgene into the genome of human hiPSCs (the *mCherry* gene, used

as a negative control). After the DT selection and cell line expansion, we tested their sensitivity to ganciclovir – a synthetic substrate for the viral TK that ultimately acts to inhibit DNA replication⁸. While the hiPSC cells expressing mCherry were insensitive to ganciclovir, the cells expressing HSV-TK did not survive the treatment (Crystal Violet staining and Cell-Titer Glo measurement, Fig. R6). This data was added to the Figure 5f,g of the main manuscript. Altogether, our data demonstrates that the Xential approach can be applied to introduce a transgene of a therapeutic relevance.

Figure R6. The *HSV-TK* gene inserted in the HBEGF locus using Xential renders hiPSCs sensitive to ganciclovir. Crystal violet staining assay (A) and CellTiter-Glo assay (B) showing viability of cells containing either the *mCherry* or the *HSV-TK* gene inserted in the HBEGF locus. Cells were treated with depicted concentration of ganciclovir or DMSO (control) for three days followed by a three-day recovery without the drug.

6 - The authors mention that one of the bottlenecks of the genome editing field is the low editing rates in some relevant cells, as T cells, especially using base editors. The authors clearly showed that they were able to increase those numbers upon selection (although modestly, 1.7 and 1.8-fold). However, new versions of base editors that perform better than their ancestors were described recently. Based on this, would the DT-selection system still outperform the new versions of ABE and CBE?

We agree with the reviewer comments. In fact, we propose that our DT selection systems can be applied to enrich edited cells regardless of version of genome editors. Reviewer's suggestion prompted us to test the latest versions of base editors with our co-selection^{4,5}.

To this end, we used plasmid transfection of ABE (ABE8e) and CBE (CBE4max) in both HEK293 and hiPSC. We observed a significant improvement of baseline editing in both cell types with the latest version of base editors compared to previous versions (Fig. 2,5; Fig. R7). In addition, DT selection enriches for editing events across all target sites with editing events ranging from 9% to 59% (Fig. R7, Supplementary Figure 3 and 7). This data demonstrates that the DT selection systems is robust, universal and compatible with novel genome editor in both cancer cells and stem cells.

Figure R7. DT co-selection increases efficiency of CBE4max and ABE8e. (a) Bar graph of co-selected cytidine base editing events at indicated loci in HEK293 with CBE4max, with or without DT selection, showing C-T conversion (%). (b) Bar graph of co-selected cytidine base editing events in hiPSC with CBE4max, with or without DT selection, showing C-T conversion (%). (c) Bar graph of co-selected adenosine base editing events in HEK293 with ABE8e, with or without DT selection, showing A-G conversion (%). (d) Bar graph of co-selected adenosine base editing events in hiPSC with ABE8e, with or without DT selection, showing A-G conversion (%). In all graphs, the values and error bars reflect mean \pm s.d. of n=3 independent biological replicates. Relative fold-changes between DT-selected and non-selected cells are indicated in the graphs. *P< 0.05, **P< 0.01, ***P<0.001, Student's paired t-test.

7 - Do the authors predict that the results of the HBEGF activity assay would apply for primary cells as well, leading the conclusion that the mutations in the HBEGF do not perturb gene function and that there's no effect on cell fitness?

We would like to thank the reviewer for this question. In our study, we presented a few line of evidences that the precise editing of the HBEGF gene does not perturb its function. (1) The HBEGF serves as a ligand for EGFR and activates downstream pathways⁹. The *in vitro* assay in cell lines showed that the E141K mutation in the HBEGF (Fig. S3B) does not compromise its signaling function. (2) Given the role of *HBEGF* in cell proliferation⁹, we examined fitness of genome-edited cells at this locus and did not observe any defects in cell growth (Fig. S1C).

To further address the reviewer's concern, we investigated the HBEGF^{E141K} mutation installed in hiPSC (Xential hiPSC) to test if it perturbs differentiation of hiPSC into the three germ layers. To this end, we differentiated the Xential hiPSCs into mesoderm, endoderm and ectoderm, and assayed the successful completion of this process by expression of the key pluripotency- and lineage-associated genes¹⁰. We did not observe any statistically significant changes between the expression of the assayed genes in the Xential hiPSC and the wild-type hiPSC (used as a control) (Fig. R8, Supplementary Figure 9). This data suggests that the Xential-modified hiPSCs successfully execute the differentiation process. Given the development of allogenic cell therapy-based medicines, our observation provides an attractive future application of the Xential technology.

Figure S9.

Figure R8. The *HBEGF*^{E141K} mutation installed in hiPSC does not perturb differentiation process. Heatmap shows the qPCR data of the expression of two lineage-specific genes assayed upon differentiation to mesoderm, endoderm and ectoderm (fold change, log₂ scale). The Xential hiPSCs were generated with Cas9/sgRNAin3/pHMEJ (Xential). The wild-type hiPSC (WT hiPSC) served as a positive control. Values were normalized to the expression of *GAPDH* and *ACTB* and set relative to undifferentiated controls (0) (data not included in the graph). Each differentiation condition was performed in triplicates (n=3) with n=2 qPCR technical repeats.

Reply references

1. Glenn, T. C. Field guide to next-generation DNA sequencers. *Mol. Ecol. Resour.* **11**, 759–769 (2011).
2. Yao, X. *et al.* Homology-mediated end joining-based targeted integration using CRISPR/Cas9. *Cell Res.* **27**, 801–814 (2017).
3. Zhang, J. P. *et al.* Efficient precise knockin with a double cut HDR donor after CRISPR/Cas9-mediated double-stranded DNA cleavage. *Genome Biol.* **18**, 35 (2017).
4. Koblan, L. W. *et al.* Improving cytidine and adenine base editors by expression optimization and ancestral reconstruction. *Nat. Biotechnol.* **36**, 843–848 (2018).
5. Richter, M. F. *et al.* Phage-assisted evolution of an adenine base editor with improved Cas domain compatibility and activity. *Nat. Biotechnol.* **38**, 883–891 (2020).
6. Bae, S., Park, J. & Kim, J. S. Cas-OFFinder: A fast and versatile algorithm that searches for potential off-target sites of Cas9 RNA-guided endonucleases. *Bioinformatics* **30**, 1473–1475 (2014).
7. Taheri-Ghahfarokhi, A. *et al.* Decoding non-random mutational signatures at Cas9 targeted sites. *Nucleic Acids Res.* **46**, 8417–8434 (2018).
8. Martin, R. M. *et al.* Improving the safety of human pluripotent stem cell therapies using genome-edited orthogonal safeguards. *Nat. Commun.* **11**, 1–14 (2020).
9. Oda, K., Matsuoka, Y., Funahashi, A. & Kitano, H. A comprehensive pathway map of epidermal growth factor receptor signaling. *Mol. Syst. Biol.* **1**, 2005.0010 (2005).
10. Tsankov, A. M. *et al.* A qPCR ScoreCard quantifies the differentiation potential of human pluripotent stem cells. *Nat. Biotechnol.* **33**, 1182–1192 (2015).

Reviewers' Comments:

Reviewer #1:

None

Reviewer #2:

Remarks to the Author:

The authors have addressed my concerns, and I would recommend this manuscript for publication in Nature Communications.